# Calibration of Low-Cost Particulate Matter Sensors PurpleAir: Model Development for Air Quality under High Relative Humidity Conditions

Martine E. Mathieu-Campbell[1], Chuqi Guo[2], Andrew Grieshop[3], Jennifer Richmond-Bryant[1,2]

[1] Center for Geospatial Analytics, North Carolina State University, Raleigh, NC 27695, USA
[2] Department of Forestry and Environmental Resources, North Carolina State University, Raleigh, NC 27695, USA
[3] Department of Civil, Construction and Environmental Engineering, North Carolina State University, Raleigh, NC 27695, USA

*Correspondence to*: Jennifer Richmond-Bryant (jrbryan3@ncsu.edu)

**Abstract.** The primary source of measurement error from the widely-used particulate matter (PM) PurpleAir sensors is ambient relative humidity (RH). Recently, the U.S. EPA developed a national correction model for $PM_{2.5}$ concentrations measured by PurpleAir sensors (Barkjohn model). However, their study included few sites in the Southeastern U.S., the most humid region of the country. To provide high-quality spatial and temporal data and inform community exposure risks in this area, our study developed and evaluated PurpleAir correction models for use in the warm-humid climate zones of the U.S. We used hourly PurpleAir data and hourly reference grade $PM_{2.5}$ data from the EPA Air Quality System database from January 2021 to August 2023. Compared with the Barkjohn model, we found improved performance metrics with error metrics decreasing by 16-23 % when applying a multi linear regression (MLR) model with RH and temperature as predictive variables. We also tested a novel semi-supervised clustering (SSC) method and found that a nonlinear effect between $PM_{2.5}$ and RH emerges around a RH of 50 % with slightly greater accuracy. Therefore, our results suggested that a clustering approach might be more accurate in high humidity conditions to capture the non-linearity associated with PM particle hygroscopic growth.

## 1 Introduction

In recent years, many communities started using low-cost particulate matter sensors to predict community exposure risks (Bi et al., 2020, 2021; Chen et al., 2017; Jiao et al., 2016; Kim et al., 2019; Kramer et al., 2023; Lu et al., 2022; Snyder et al., 2013; Stavroulas et al., 2020), since short-term and long-term exposures to particulate matter (PM) with an aerodynamic diameter of 2.5 µm or smaller ($PM_{2.5}$) are associated with several adverse health effects (Brook et al., 2010; Chen et al., 2016; Cohen et al., 2017; Health Effects Institute, 2020; Landrigan et al., 2018; Olstrup et al., 2019; Pope and Dockery, 2006). These

low-cost sensors have been used to inform exposure risks in different applications including environmental justice (Kramer et al., 2023; Lu et al., 2022), wildfire exposure (Kramer et al., 2023), traffic-related exposure (Lu et al., 2022), and indoor exposure (Bi et al., 2021; Lu et al., 2022). The dense monitoring network enabled by deploying low-cost sensors provides the potential to understand the $PM_{2.5}$ exposure risk at a higher spatial and temporal resolution than the established regulatory air quality monitoring system. Federal Reference Method or Federal Equivalence Method (FRM/FEM) monitors tend to be sparsely sited due to the cost and complexity of this instrumentation.

Several studies have evaluated the performance of low-cost PM sensors for different sources and meteorological conditions, with bias and low precision reported in several cases (Ardon-Dryer et al., 2020; Barkjohn et al., 2021; Bi et al., 2020, 2021; He et al., 2020; Holder et al., 2020; Jayaratne et al., 2018; Kelly et al., 2017; Kim et al., 2019; Magi et al., 2020; Malings et al., 2020; Sayahi et al., 2019; Stavroulas et al., 2020; Tryner et al., 2020; Wallace et al., 2021). A study conducted in 2016 (AQ-SPEC. http://www.aqmd.gov/docs/default-source/aq-spec/field-evalua tions/purpleair—field-evaluation.pdf. Accessed on 02/20/2024.) to evaluate low-cost $PM_{2.5}$ sensors showed an overall good agreement between PurpleAir PM sensors and two reference monitors with $R^2$ of 78 % and 90 % (AQ-SPEC, 2016). However, an overestimation of 40 % was found for PurpleAir $PM_{2.5}$ concentrations compared with the reference monitors (AQ-SPEC. http://www.aqmd.gov/docs/default-source/aq-spec/field-evalua tions/purpleair—field-evaluation.pdf. Accessed on 02/20/2024.; Wallace et al., 2021). Humidity has been documented as an important parameter that could greatly reduce the performance of low-cost sensors (Rueda et al., 2023; Wallace et al., 2021; Zusman et al., 2020). Most low-cost PM sensors, including the PurpleAir sensor, utilize optical sensors based on the light-scattering principle to estimate PM mass concentration. Thus, they are subject to measurement errors from various factors, including particle size, composition, optical properties, and interactions of particles with atmospheric water vapor (Hagan and Kroll, 2020; Rueda et al., 2023; Zheng et al., 2018; Zusman et al., 2020). In a high humidity environment, accurate detection of particle size and concentration may be affected by hygroscopic growth of particles (Carrico et al., 2010; Chen et al., 2022; Healy et al., 2014; Jamriska et al., 2008; Wallace et al., 2021). Water vapor may also damage the circuitry of the sensors (Jamriska et al., 2008; Wallace et al., 2021). Relative Humidity (RH) has therefore been confirmed to be a primary source of measurement error that requires concentration correction in low-cost PM sensors (Barkjohn et al., 2021; Sayahi et al., 2019; Wallace et al., 2021; Zusman et al., 2020).

The PurpleAir PM sensor is one of the most widely used low-cost PM sensors (Bi et al., 2021; Wallace et al., 2021). As of April 2022, there were more than 30,000 networked PurpleAir sensors, providing geolocated real-time air quality information (https://www2.purpleair.com, https://www.airnow.gov). Recently, the U.S. Environmental Protection Agency (EPA), after an evaluation of the sensors, developed a national correction model for PurpleAir sensors (Barkjohn et al., 2021). However, this evaluation included few sites in the Southeastern U.S. (Barkjohn et al., 2021).The study covered 16 states using 39 sites selected according to their collocation with an FRM/FEM monitor. In this study, the Southeastern U.S., the most humid region of the U.S., characterized by a humid subtropical climate (Konrad et al., 2013), was represented by only 5 sites and

encompassed 4 states. The EPA correction model used multi-linear regression (MLR) (Barkjohn et al., 2021). Some recent studies used model-based clusters (MBC) to improve performance metrics compared with their MLR models. McFarlane et al. (2021) and Raheja et al. (2023) applied a Gaussian Mixture Regression (GMR) bias correction model to $PM_{2.5}$ PurpleAir sensors in Accra, Ghana. The GMR-based model developed by McFarlane et al. (2021) used daily data from one PurpleAir sensor collocated with one Met One Beta Attenuation Monitor 1020 from March 2020 to March 2021. Raheja et al. (2023) used 3 different brands of low-cost sensors including PurpleAir PA-II collocated with a Teledyne T640 as the reference grade monitor at the University of Ghana, in Accra, Ghana, from May to September 2021. However, a GMR-based model is not transferable to new settings (Raheja et al., 2023), since the regression function in a GMR is derived from input from modeling the joint probability distribution of the data (Maugis et al., 2009; McFarlane et al., 2021; Shi and Choi, 2011). The model is not flexible enough to handle differences in proportions of the input variables observed at different locations.

The objective of this study is to develop and evaluate PurpleAir bias correction models for use in the warm humid climate zones (2A and 3A) of the U.S. (Antonopoulos et al., 2022). First, we tested an MLR with different combinations of predictive variables. To avoid the transferability constraints observed for the GMR, our study then tested a novel semi-supervised clustering method. We used PurpleAir data and the FRM/FEM $PM_{2.5}$ data from the EPA Air Quality System (AQS) database from January 2021 to August 2023. We tested new correction models developed for the high-humidity Southeastern region of the country and compared them with the EPA nationwide PurpleAir data correction model proposed by Barkjohn et al. (2021).

## 2 Methods

### 2.1 Study area

The study area includes the "warm-humid and moist" climate zone of the United States, as defined by the International Energy Conservation Code (EICC) in 2021. The 2021 EICC identifies the appropriate climate zone designation for each county in the U.S. (Antonopoulos et al., 2022). The climate zone map comprises eight regions at county level, with seven represented in the continental U.S. (Antonopoulos et al., 2022; Chapter 3, General requirements, 2021 International Energy Conservation Code (IECC)).The thermal climate zones are based on meteorological parameters (designated as 1 to 8) including precipitation, temperature, and humidity and a moisture regime (designated as A, B, C for Humid/Moist, Dry, and Marine respectively). The thermal climate is determined using the heating degree-days (HDD) and cooling degree-days (CDD), and the moisture regime is based on monthly average temperature and precipitation (Antonopoulos et al., 2022; Chapter 3, General requirements, 2021 International Energy Conservation Code (IECC)).

The study area was composed of climate zones and moisture regimes 2A and 3A. The "warm-humid" climate zone designation corresponds to a specific area of the climate zone map that includes Zones 2A and 3A (Fig. 1). Zone 1A is excluded, given that its tropical characteristics are sufficiently different from most of the Southeast. A warm and humid climate is characterized

by high levels of humidity and high temperatures throughout the year and receives more than 20 inches (50 cm) of precipitation per year (Baechler et al., 2015). This area presents a state average annual humidity varying between 65.5 and 74.0 % and an

average temperature per state varying between 55.1 and 70.7° F. These 12 states have the 12 highest annual average dewpoint temperatures in the Continental U.S.

The study area includes 799 counties distributed into the 12 states as shown in the table in Fig. 1. Excepting Kentucky, all of the Southeastern U.S. states are partially or entirely characterized by a warm-humid climate zone and included in our study

area. The high humidity condition in this part of the U.S. might affect particle composition and size distribution due to water uptake (Hagan and Kroll, 2020; Jaffe et al., 2023; Patel et al., 2024; Rueda et al., 2023). A study conducted in 2018 (Carlton et al., 2018) found large contributions (50 %) to $PM_{2.5}$ from biogenic secondary organic aerosols (BSOA) in the Southeast U.S. region compared with the rest of the country. The elevated BSOA are attributed to heavily forested areas and large urban areas in the region (Carlton et al., 2018; U.S. EPA, 2018).

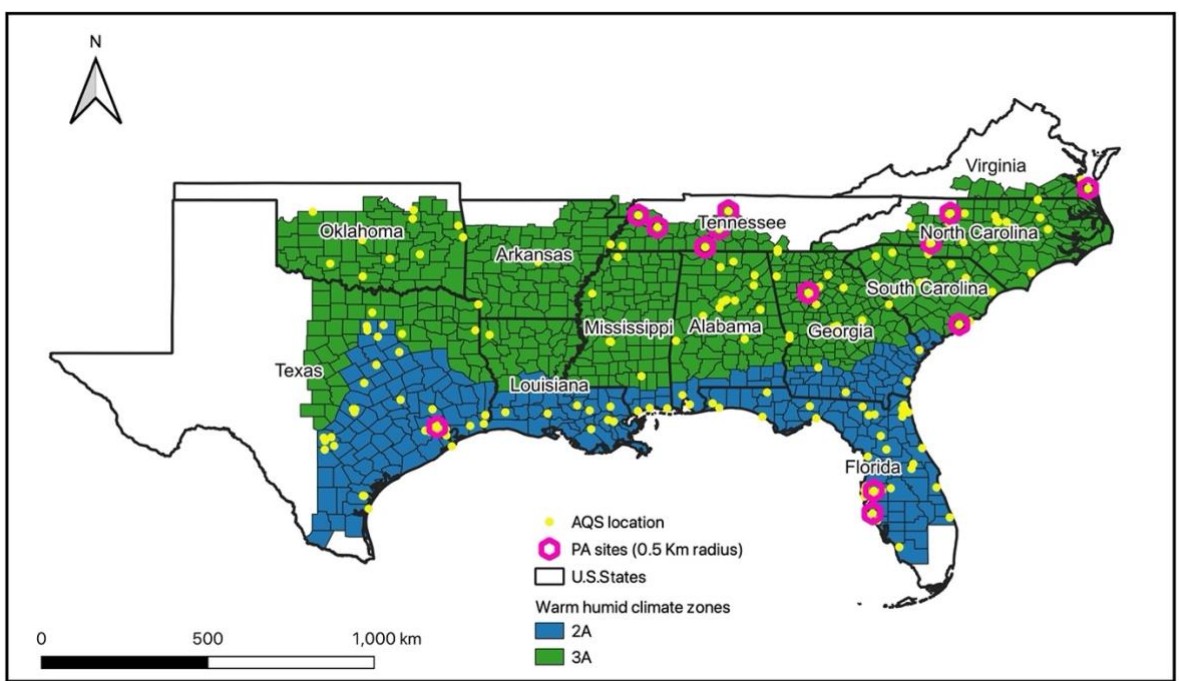


**Figure 1: Study area showing the warm humid climate zones classification. The map also shows the distribution of available AQS monitors and the distribution of the PurpleAir sensors (PA sites) located at 0.5-km radius of an AQS monitor.**

### 2.2 Data collection

The PurpleAir (PA-II-SD) contains two Plantower PMS5003 laser scattering particle sensors, a pressure-temperature-humidity

sensor (BME280), and a Wi-Fi module (Magi et al., 2020). $PM_{2.5}$ data from the PurpleAir sensors were obtained from the PurpleAir data repository (API PurpleAir. https://api.purpleair.com), and $PM_{2.5}$ data from the State and Local Air Monitoring

System (SLAMS) were retrieved from the U.S. Environmental Protection Agency (EPA) Air Quality System (AQS) (U.S. EPA, 2023) for the period from 1 January 2021 to 28 August 2023 using their respective application programming interfaces (API). To obtain data for the study area, we used a bounding box (-100.01º W, -75.50º E and 25.81º S, 37.01º N) to find all

outdoor sensors available for this geographical area. We identified 997 available sensors. We used the $PM_{2.5}$ dataset related to a standard environment, which was reported in the PurpleAir output as cf_1 (correction factor = 1). This represents a more appropriate raw measurement of PM concentrations without any nonlinear transformation (McFarlane et al., 2021), and has been used for several other studies (Barkjohn et al., 2021; Raheja et al., 2023; Tryner et al., 2020; Wallace et al., 2021). Hourly average $PM_{2.5}$ concentrations were downloaded for both PurpleAir sensors and AQS monitors.


SLAMS data are collected by local, state, and tribal government agencies and made available via the AirNow API (U.S. EPA, 2023a). To ensure data accuracy, AQS data are collected by FRM or FEM (U.S. EPA, 2023b). These methods are primarily maintained to evaluate compliance with the National Ambient Air Quality Standards (NAAQS), although the data are often used for air pollution exposure and epidemiology research. We identified 181 FEM or FRM monitors in our study area.

**2.3 Selection of PurpleAir sensors and data quality control criteria**

We selected PurpleAir sensors within fixed radii of each FRM or FEM monitor. The R Statistical Software (version R 4.3.1) was employed for data selection, data quality control, and statistical modeling. We identified outdoor PurpleAir sensors within 2.0, 1.0, and 0.5 km of each FRM or FEM monitor. When a PurpleAir sensor fell within the buffers of 2 or more AQS monitors, the shorter distance to the AQS buffer centroid was applied to ensure better spatial join accuracy.


We applied a series of data exclusion criteria for quality control. First, we used a detection limit of 1.5 μg m⁻³ for the PurpleAir data. This value is equivalent to the average of the values reported by Tryner et al. (2020) and Wallace et al. (2021) for the cf_1 data series. We also excluded all $PM_{2.5}$ data points that were greater than 1000 μg m⁻³. Then, we applied data exclusion criteria to clean the PurpleAir data based on agreement between the concentrations reported for the two Plantower PMS5003

sensors provided in the PurpleAir housing, labeled arbitrarily as Channels A and B. We considered low and high concentrations separately. For low $PM_{2.5}$ concentrations (less than or equal to 25 μg m⁻³), we removed observations where the concentration difference between Channels A and B was greater than 5 μg m⁻³ and the percent error deviation was greater than 20 %. For high $PM_{2.5}$ concentrations (greater than 25 μg m⁻³), we removed data records when the percent error deviation between Channels A and B was greater than 20 %. Similar cleaning criteria were used for quality assurance by Barkjohn et al.

(2021) and Tryner et al. (2020), where data with a difference between Channels A and B less than 5 μg m⁻³ for low $PM_{2.5}$ concentration were considered valid. Bi et al. (2020) removed data with the 5 % largest percent error difference between Channels A and B. Additionally, Barkjohn et al. (2021) excluded data points where Channels A and B that deviated by more than 61 %. However, we decided to employ a more stringent criterion for our high concentration data records (20 % deviation)

considering that our study only included reported PurpleAir data available via the API and only for one region of the United

States. Following data cleaning, the final PurpleAir concentration ($C_{PA}$) dataset used in our study was obtained by averaging Channels A and B and included only hourly average PurpleAir data points that had a spatial (within the calculated radius) correspondence to hourly FEM[1] concentration ($C_{AQS}$) data. Missing $C_{AQS}$ data points were excluded before applying the radius-related spatial join.

To ensure data quality, the relative humidity measured by the BME280 sensor within the PurpleAir housing was evaluated. We compared hourly RH from the PurpleAir with the corresponding hourly RH from the National Oceanic and Atmospheric Administration (NOAA) Integrated Surface Database (ISD). The NOAA data were downloaded using the R package *worldmet* (worldmet: Import Surface Meteorological Data from NOAA ISD). The nearest NOAA station to each PurpleAir sensor was considered for the comparison. The average distance between a NOAA station and a PurpleAir sensor was approximately 10

miles with a minimum of 1.65 miles and a maximum of 25.50 miles. All PurpleAir sensors that presented a correlation of less than 0.80 with the corresponding RH from NOAA were excluded.

## 2.4 Model correction

### 2.4.1 Model inputs

Because measurement errors are related to water uptake by particles (Hagan and Kroll, 2020; Rueda et al., 2023; Wallace et

al., 2021), temperature (T) and relative humidity (RH) are the most commonly found bias correction parameters in the literature (Ardon-Dryer et al., 2020; Bi et al., 2020; Magi et al., 2020; Malings et al., 2020; Wallace et al., 2021) for the PurpleAir. Thus, our meteorological data (hourly T, hourly RH) were taken from the PurpleAir sensor, similar to the analysis conducted by Barkjohn et al. (2021). Barkjohn et al. (2021) included dewpoint temperature (DP) in addition to T and RH as input predictors in their modeling process. However, DP was excluded as a predictor in our study. DP exhibited collinearity with both RH and

T when testing for variance inflation factor. In fact, a high correlation of 95 % was found between DP and T. Therefore, including it would inflate the goodness of fit of the model. This result is not surprising considering the interdependent atmospheric thermodynamic relationship of DP with RH and T. For data quality assurance, we only included data records within a range of 0-130 °F for T and 0-100 % for RH, respectively. Similar quality assurance criteria were employed by Wallace et al. (2021) where data records with abnormal temperature and relative humidity measurements were removed.


The final dataset used for our model calibration included $C_{PA}$, $C_{AQS}$, RH, and T. We tested several multilinear regression models, and we defined a supervised clustering approach.

---

[1] The AQS reference monitors used in our study were FEM monitors.

### 2.4.2 Multilinear Regression

Our study tested five Multilinear Regression (MLR) models (Equations 1-5) including the model proposed by Barkjohn et al.
(2021) (Model Bj). Based on the evaluated predictors, we developed Model 1-4. The four proposed models and the Barkjohn model were structured as follows:

Model 1: $C_{AQS} = \beta_0 + \beta_1 C_{PA} + \varepsilon$ (1)

Model 2: $C_{AQS} = \beta_0 + \beta_1 C_{PA} + \beta_2 RH + \varepsilon$ (2)

Model 3: $C_{AQS} = \beta_0 + \beta_1 C_{PA} + \beta_2 T + \varepsilon$ (3)

Model 4: $C_{AQS} = \beta_0 + \beta_1 C_{PA} + \beta_2 RH + \beta_3 T + \varepsilon$ (4)

Model Bj: $C_{AQS}^2 = 5.72 + 0.524 * C_{PA} - 0.0852 * RH$ (5)

For each model, $\beta_0$ represents the intercept, $\beta_1$ - $\beta_3$ are the coefficient of the predictors $C_{PA}$, RH and T respectively, and $\varepsilon$ is the error term.

### 2.4.3 Semi-supervised Clustering

Alternative bias correction methods to MLR have been developed (Bi et al., 2020; McFarlane et al., 2021; Raheja et al., 2023) to capture complex nonlinear hygroscopic growth of particles (Hagan & Kroll, 2020; bark et al., 2023). Some of these alternative techniques include model-based clusters (MBC) (McFarlane et al., 2021; Raheja et al., 2023). An MBC assumes that the data are composed of more than one subpopulations (Raftery and Dean, 2006). The influence of RH on PurpleAir $PM_{2.5}$ measurements, specifically at high ambient RH (Wallace et al., 2021), may be non-linear, suggesting formation of subgroups in our dataset. Therefore, our study tested a semi-supervised clustering (SSC) approach that combines unsupervised and supervised clustering processes to develop a non-linear MBC (Raftery and Dean, 2006). Before implementing the SSC, we carried out two pre-processing steps. The first pre-processing step consisted of finding the optimal predictors for the clusters by applying a Gaussian Mixture Model (GMM) variable selection function (forward-backward) for MBC (Raftery and Dean, 2006). The GMM variable selection process uses the expectation-maximization (EM) algorithm to determine the maximum likelihood estimate for GMM (Raftery and Dean, 2006). The optimal variables are then selected using the Bayesian information criterion (BIC). The list of potential variables included RH and T (the variable DP was excluded in this process because of multicollinearity with RH and T). The second pre-processing step was to determine the optimal number of clusters. For this, we used a combination of 26 clustering methods via the NbClust R package (Boehmke and Greenwell, 2019; Charrad et al., 2014). Knowing the optimal variable predictors and the optimal number of clusters, we initiated the unsupervised portion of our SSC using the K-means clustering algorithm. K-means, one of the most commonly employed clustering methods, is an

---

[2] $C_{AQS}$ here represents the reference $PM_{2.5}$ monitors used in Barkjohn et al. (2021).

unsupervised machine learning partitioning distance-based algorithm that computes the total within-cluster variation as the sum of squared (SS) Euclidian distances between the centroid of a cluster $C_k$ and an observation $x_i$ based on the Hartigan-Wong algorithm (Hartigan and Wong, 1979; Yuan and Yang, 2019). Last, we applied a supervised clustering process built upon the results obtained for the unsupervised clustering approach. The supervised process allowed for distribution of the dataset within well-defined subsets. For each subset of the dataset associated with a cluster, an MLR was developed, defining a non-linear MBC (Equation 6).

$$y = \begin{cases} \beta_0 + \beta_1 x_{i1 \in C_1} + \cdots + \beta_p x_{ip \in C_1} + \in \\ \beta_0 + \beta_1 x_{i1 \in C_k} + \cdots + \beta_p x_{ip \in C_k} + \in \end{cases} \qquad (6)$$

where $C_k$ is the number k of clusters regrouping $x_i$ observations for each p explanatory variable.

### 2.4.4 Model validation

For each of the evaluated models, the coefficient of determination, $R^2$, was calculated to understand how well the regression model performs with the selected predictors. The predictive performance of each model was evaluated by estimating Root Mean Square Error (RMSE) and Mean Absolute Error (MAE). RMSE is the standard deviation of the prediction errors. MAE measures the mean absolute difference between the predicted values and the actual values in a dataset. Standard deviation (SD), $R^2$ and RMSE are EPA's recommended performance metrics to evaluate a sensor's precision, linearity, and uncertainty, respectively (Duvall et al., 2021). We compared EPA's target value for SD, which refers to collocated identical sensors, with the estimated mean deviation or MAE for each paired observation of $C_{AQS}$ and $C_{PA}$.

### 2.4.5 Cross-validation

Building the correction model based on the full dataset could overfit the model (Barkjohn et al., 2021). Therefore, we used leave-one-group-out cross-validation (LOGOCV) methods to evaluate how the model performs for an independent test dataset. LOGOCV involves splitting the dataset into specific or random groups, then predicting each group as testing data with the other groups used for training. We used an automatic LOGOCV, in which a random set of training data was composed to predict $PM_{2.5}$ concentrations at each iteration. An 80/20 ratio was defined between the training and test groups with 25 iterations. Then, we applied a leave-one-state-out cross-validation (LOSOCV) that involves splitting the dataset into specific states to evaluate the performance of the model. In our LOSOCV, every U.S. state was left out successively and used in a validation test, while the remaining states were used to train the model. We used $R^2$, RMSE, and MAE as performance metrics to evaluate the cross-validation results.

### 2.4.6 Sensitivity analysis

Sensitivity analyses were performed to determine how predictions of $PM_{2.5}$ concentrations would vary under different temporal resolution. The sensitivity analysis applied the models, developed from hourly data at 0.5-km, 1.0-km, and 2.0-km buffers, to daily averaged data for the same buffers. We applied a completeness criterion of 90 %, or 21 hours, following Barkjohn et al. (2021)

### 3 Results and Discussion

After applying all the quality assurance (QA) criteria to the raw datasets, we obtained 159,648 observations (18 PurpleAir sites), 238,047 observations (28 PurpleAir sites), and 394,010 observations (50 PurpleAir sites) for buffers of 0.5 km, 1.0 km, and 2.0 km respectively, all at hourly temporal resolution. The QA process removed about 22 % (Table S1) of the raw data, with data from 3 PurpleAir sites completely removed for the 0.5-km radius because RH from the humidity sensors correlated poorly with RH reported by NOAA stations (Fig. S1). We found that two of these same 3 PurpleAir sites exhibited poor

correlation for temperature as well. Moreover, the slope of the linear regression estimated for each PurpleAir sensor (Fig. 1) shows that RH from these 3 PurpleAir sites exhibited the larger bias metrics. All 18 retained PurpleAir sites presented RH data that strongly correlated with NOAA stations (88-96 %), with 16 of them presenting an R equal or greater than 90 % (Fig. S1). As reported by recent studies (Barkjohn et al., 2022; Giordano et al., 2021; Magi et al., 2020; Tryner et al., 2020), PurpleAir sensors tend to report dryer humidity measurements than ambient conditions. The comparison of our PurpleAir sensors with

NOAA stations showed that each of the 18 retained PurpleAir sites reported lower humidity measurements than their corresponding NOAA station. They presented a negative difference in RH varying between 10-20 %, with uncertainty increasing with increased RH (Fig. S2). In addition to the 3 PurpleAir sites removed for the 0.5-km radius, 1 and 2 additional PurpleAir sites were removed for the 1.0 km and 2.0 km buffers, respectively. We did not detect any additional instrument error for temperature. Most of the retained PurpleAir sites had a strong correlation of 95-99 % for temperature with NOAA

stations.

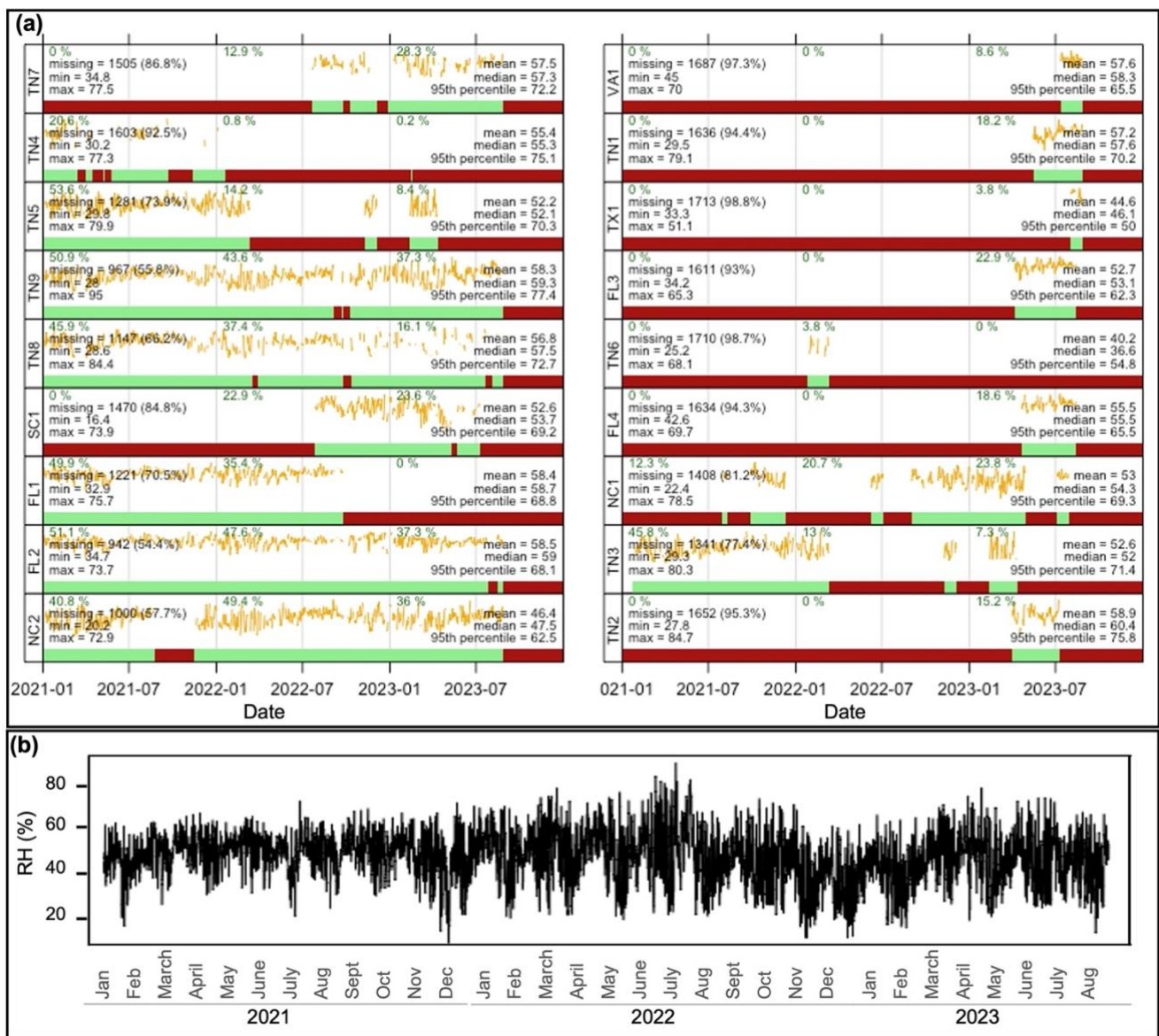

**Figure 2: (a) Summary statistics and time series (yellow lines) of daily average RH for each PurpleAir site showing the presence of data (green) and missing data (red). The y axis represents RH scaled from zero to the maximum daily value. The percentage of data captured per year is also provided. (b) Time series of daily average RH for the entire dataset with a SD of 10.56 %.**


Summary statistics were explored to describe the main characteristics of our datasets (Fig. 2 and 3). Meteorological parameters for our three buffers (0.5 km, 1.0 km and 2.0 km) exhibit roughly the same distribution (Fig. S3). Further evaluation of our 0.5-km radius dataset revealed that 63 % of the hourly data for RH are greater than 50 % with temperatures varying between -17.13 and 38.83 °C. RH for the 0.5-km radius dataset showed some monthly seasonality (Fig. 2B). However, independent of

the number of months of data reported by a PurpleAir sensor, the distribution of RH is relatively consistent for individual

PurpleAir sites (Fig. 2A). For this same radius, the number of complete months of data per PurpleAir sensor varied from approximately 1 to 29 months, with 11 sensors covering at least 10 months of hourly data (Fig. 2A).

For the PM$_{2.5}$ concentration data, Fig. 3 displays the mean and SD for the C$_{AQS}$ and C$_{PA}$ data for all three analyzed buffers. The
Pearson correlation (R), R$^2$, RMSE and MAE between C$_{AQS}$ and C$_{PA}$ before fitting any model were also estimated for each radius (Fig. 3). All of the metrics, R$^2$, MAE and RMSE exceeded the target values[2] (R$^2 \geq 70$ %, SD $\leq 5$ µg m$^{-3}$ and RMSE $\leq 7$ µg m$^{-3}$) recommended by EPA (Duvall et al., 2021). Raw C$_{PA}$ presented greater magnitude and variability than C$_{AQS}$ (Fig. 3). The performance metrics (Tables 1 and 2, Tables S2-S5) indicated less error with successively smaller buffer size, which suggests that model fit improves with decreased distance between the AQS monitors and PurpleAir sensors. The distance
factor might be attributed to spatial variability between AQS monitors and PurpleAir sensors and the effect of various potential PM sources around the air monitors. Therefore, we present only the results for the 0.5-km buffer analysis. Tables S2-S5 contain the results for the 1.0-km and 2.0-km buffers, respectively. Wallace et al. (2021) and Bi et al. (2021) also used a 0.5-km buffer around the AQS monitors in their low-cost sensor data calibration studies.

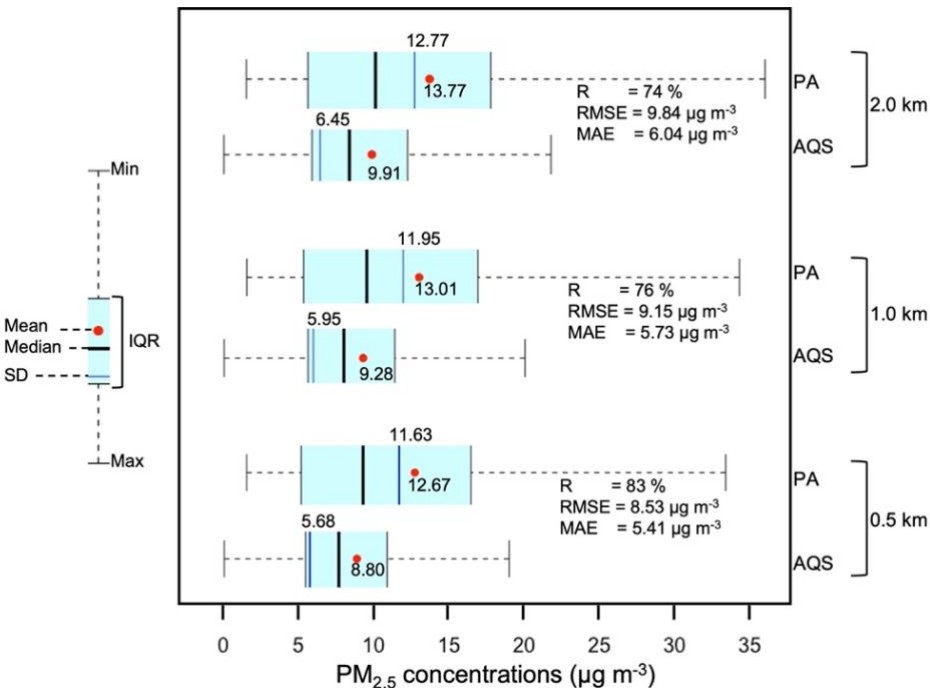


**Figure 3: Descriptive and error metrics for C$_{AQS}$ and raw C$_{PA}$ for PurpleAir sensors within a 0.5-km, 1.0-km and 2.0-km radii of each FRM or FEM monitor.**

## 3.1 MLR Bias-Correction Model

The bias-correction models, including the Barkjohn model (2021), and their performance metrics are presented in Table 1. All four MLR-fitted models exhibited an average concentration of 8.80 μg m$^{-3}$, with a SD varying between 4.71- 4.84 μg m$^{-3}$. The Barkjohn model had a mean of 7.67 μg m$^{-3}$ and a SD of 6.08 μg m$^{-3}$. RMSE and MAE, which summarize the error on hourly PM$_{2.5}$ averages, exhibited relatively low values for the four fitted models when we consider the average C$_{AQS}$ in the dataset and its SD, and the EPA's target value[3] ($\leq 7$ μg m$^{-3}$) for RMSE. Our dataset illustrates improved predictive performance for our four MLR-fitted models compared with the Barkjohn model (Table 1). The Barkjohn model presented a higher R$^2$, as a measure of the goodness of fit, than Model 1, however Model 1 is improved with respect to all error metrics. The Barkjohn model resulted in a higher MAE than the four models developed for this study.  The best model fit was observed for Model 4, incorporating C$_{PA}$, T, and RH, with substantially better prediction performance metrics compared with the other models (Table 1). The model would, however, be further improved with use of newer PurpleAir sensors because, over time, the quality of the sensors degrades. This is particularly true in the hot and humid climate zone (deSouza et al., 2023). Similarly, the presence of Teledyne T640s among our AQS monitors may have affected the performance of our models since positive bias of approximately 20 % has been reported with T640s compared with other FEM or FRM monitors (U.S. EPA, 2024). Additionally, a study conducted by Searle et al. (2023) found that 12.9 % of the sensors deployed by PurpleAir between June 2021 and May 2023 reported negative bias of approximatively 3 μg m$^{-3}$. These PurpleAir sensors, specifically deployed between June 2021 and January 2022 and between March to May 2023, used an alternative Plantower PMS5003 that affected the reported particle size distributions and concentrations (Searle et al., 2023). Based on the technique developed by Searle et al. (2023) to identify PMS5003 sensors, we estimated that only one of our sensors (sensor ID: 116559), representing 0.62 % of our data, fell into this category. This may have a slight effect on the performance of our models. Furthermore, unlike our fitted models, Model Bj applied to our dataset displayed some negative values. Model 2 was similar in structure to the selected model from Barkjohn et al. (2021), with C$_{PA}$ and RH as predictors. All predictors for every model were statistically significant. Validation testing using the LOGOCV (Table S6) presented nearly identical results to models using the entire dataset, building confidence in the models. The LOSOCV resulted in a RMSE and a MAE of 3.32 μg m$^{-3}$ and 2.29 μg m$^{-3}$ respectively for Model 4. These values were higher than those for the LOGOCV process, which is not surprising considering the variability between states.

Our findings align with some previous low-cost sensor data calibration work (Barkjohn et al., 2021; Magi et al., 2020; Zheng et al., 2018), where relatively simple calibration models provided reasonable bias correction. Zheng et al. (2018), evaluating the performance of Plantower PMS3003, which is similar to the PM$_{2.5}$ sensor used in PurpleAir, found an R$^2$ value of 66 % for a 1-h averaging period after applying an MLR calibration equation to compare three Plantower sensors against each other and a co-located reference monitor over a period of 30 days. A study conducted by Magi et al. (2020), involving a sixteen-

---

[3] The EPA's target values were estimated for 24h average data.

month PurpleAir PM$_{2.5}$ data collection in an urban setting in Charlotte, North Carolina, resulted in R$^2$ of 60 % for an MLR
including C$_{PA}$, RH and T. Barkjohn et al. (2021) estimated an RMSE of 3 µg m$^{-3}$ (no decimal specified) when fitting a model
with RH for a mean concentration of 9 µg m$^{-3}$ for FRM or FEM monitors. Moreover, the negative coefficient obtained for RH
for Model 2 and Model 4 is not surprising considering that high RH can lead to hygroscopic growth of the particles, and
therefore cause uncertainties and overestimation in PurpleAir PM$_{2.5}$ concentration readings (Bi et al., 2021; Wallace et al.,
2021). The model developed by Barkjohn et al. (2021), as well as the MLR model developed by Raheja et al. (2023) using
data in Accra, Ghana, had a negative coefficient for RH.

Table 1: MLR model development (model fit using hourly data) and application of the hourly model to daily data.

| Parameters | | Model fit with hourly data | | | | Model fit to daily data | | | |
|---|---|---|---|---|---|---|---|---|---|
| Models | | R$^2$ (%) | RMSE (µg m$^{-3}$) | MAE (µg m$^{-3}$) | R (%) | R$^2$ (%) | RMSE (µg m$^{-3}$) | MAE (µg m$^{-3}$) | R (%) |
| Model 1 | 3.6667550 + 0.4053418PA$_i$ | 69 | 3.16 | 2.13 | 83 | 76 | 2.39 | 1.67 | 87 |
| Model 2 | 6.3384228 + 0.4143437PA$_i$ - 0.0506037RH$_i$ | 71 | 3.05 | 2.05 | 84 | 76 | 2.35 | 1.64 | 87 |
| Model 3 | 1.7642336 + 0.4109897PA$_i$ + 0.0847196T$_i$ | 71 | 3.04 | 2.06 | 84 | 77 | 2.32 | 1.67 | 88 |
| Model 4 | 4.3295358 + 0.4182906PA$_i$ - 0.0445768RH$_i$ + 0.0752867T$_i$ | 73 | 2.96 | 1.99 | 85 | 79 | 2.24 | 1.59 | 89 |
| Model Bj | 5.72 + 0.524PA$_i$ - 0.0852RH$_i$ | 71 | 3.52 | 2.51 | 84 | 76 | 2.76 | 2.06 | 87 |

Following removal of datapoints that did not fit the QA criteria, the 0.5-km daily dataset included 5,666 observations for the
same 18 sensors when applying the hourly model to daily data. These produced a substantial improvement in the performance
metrics compared with those of the hourly models (Table 1). Model 4 presented better performance metrics compared to the
other models (Table 1). Figure 4 shows the correlation between the predicted C$_{PA}$ and C$_{AQS}$ for Model 4 and Model Bj along
with the distribution of RH. The model developed by Barkjohn et al. (2021) used only daily averaged data, thus, it was directly
comparable with our application of the model to daily data. An aggregate of datapoints can be seen on the left-hand side of the
correlation plots (Fig. 4) to deviate from the model fit line. These data probably influenced the performance metrics of the
models. An evaluation of Model Bj applied to our warm-humid climate zone daily PurpleAir datasets revealed substantially
higher error metrics than the other models (Table 1).

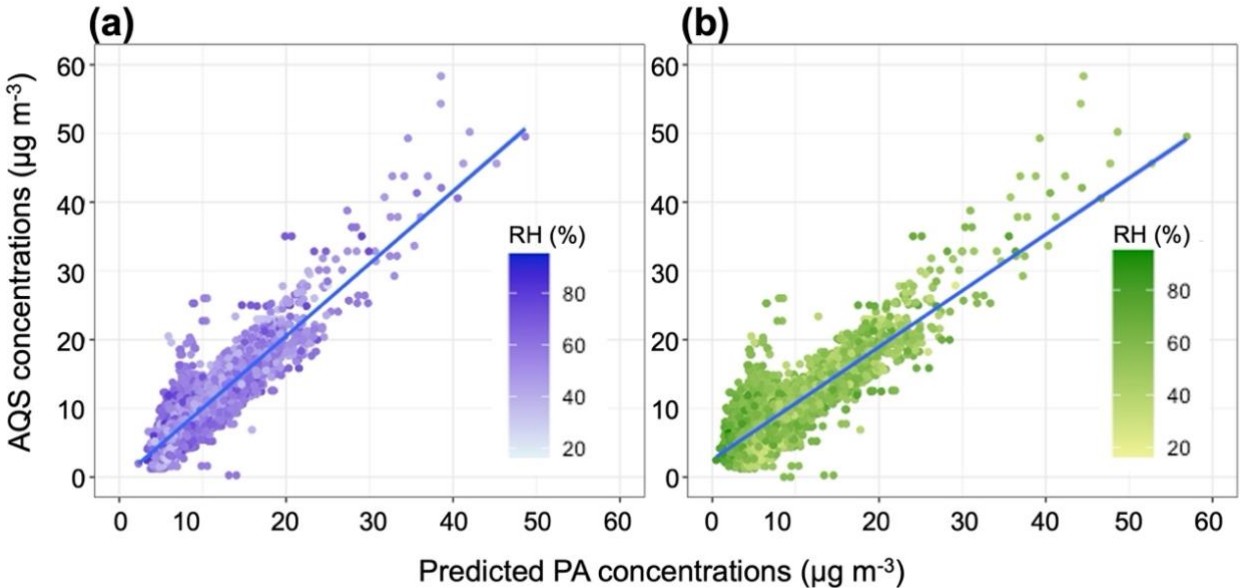

Figure 4: Positive linear correlation between daily AQS and daily predicted PM$_{2.5}$ concentrations with RH distribution (a) AQS and predicted PM$_{2.5}$ concentrations using Model 4 of the MLR process shown in purple (b) AQS and predicted PM$_{2.5}$ concentrations using the Barkjohn model shown in green.

### 3.2 SSC Model Predictions

The SSC model included the same predictors as Model 4 (C$_{PA,}$ RH and T) as the best MLR model obtained. The GMM process, discerning complex relationships between variables, found that RH and T are optimal predictors to use in the clustering process. Among the twenty-six indices evaluated, we found that eight of them proposed k=2 as the optimal number of clusters (Table S9). Thus, we set k=2 clusters for the unsupervised aspect of our SSC process. Figure 5A shows the k clusters result for the silhouette algorithm, which is based on two factors, cohesion (similarity between the object and the cluster) and separation (comparison with other clusters) (Yuan and Yang, 2019). The unsupervised clustering suggested a distribution of the dataset into two well-defined clusters based on the RH predictor (Fig. 5B). For T, the same range of values was found within each defined cluster. RH, being the most important variable that determined the clustering subdivision (Fig. 5B), therefore, we considered only RH for the cluster subdivision and then we applied the supervised phase of the SSC process to adjust the random subdivision of the clusters and eliminate overlaps. The two clusters were RH $\leq$ 50 % (Cluster 1) and RH > 50 % (Cluster 2) (Table 2). This result aligns with Wallace et al. (2021), showing that the nonlinear effect between PM$_{2.5}$ and RH emerges around a RH of 50 %, similar to our cluster division (Fig. S4).

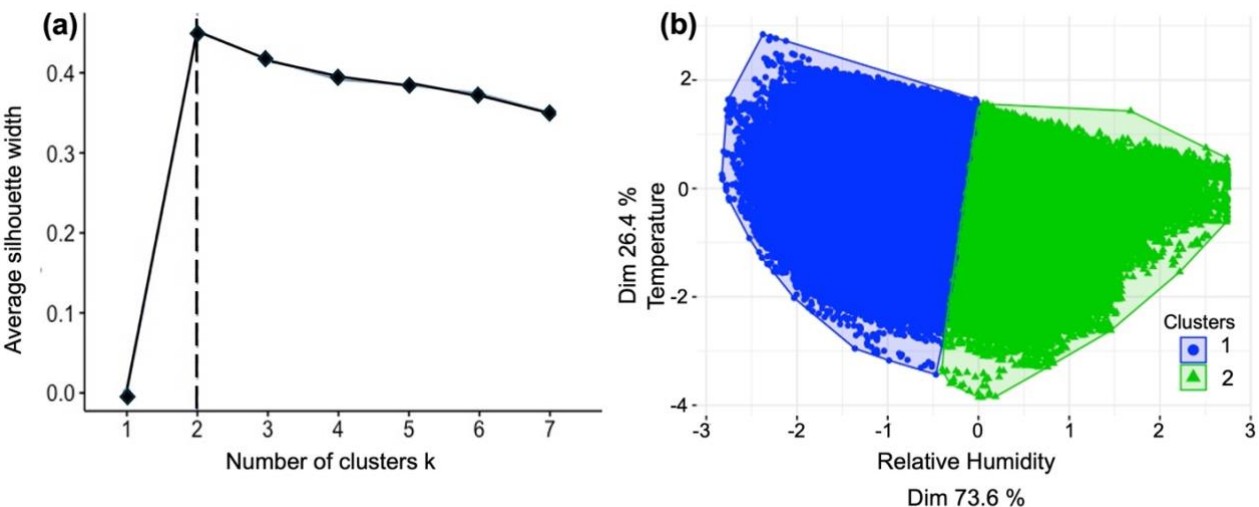

**Figure 5: Unsupervised clustering results: (a) Number of clusters k using the silhouette algorithm; (b) Clustering subsets based on RH and T showing that RH has a greater influence in the process. The axis values correspond to covariance, and the dimensionality corresponds to how much of each variable participated in the clustering process.**

The SSC approach provides improved model fits compared with the MLR models for our hourly data. Table 2 presents the modeling results of the RH-based semi-supervised clustering process. The difference between the two models resides primarily in their intercepts and their RH coefficients (Table 2). The RH factor is 10 times greater in Cluster 2 than Cluster 1, and the intercept of Cluster 2 is about 5.5 μg m$^{-3}$ greater than Cluster 1. All predictors were statistically significant. Models from both clusters are within the range of the EPA's target values for linearity and error performance metrics (Table 2). Except for MAE

that is much lower for Cluster 1, the Cluster 2 model presented better performance metrics compared with the Cluster 1 model (Table 2). Compared with Model 4 from the MLR models, results from Cluster 1 showed equal RMSE and a very low MAE, while estimated metrics from Cluster 2 are greatly improved with the exception of MAE (Table 2). The combined predicted PurpleAir concentrations from the two SSC clusters resulted in an RMSE of 2.94 μg m$^{-3}$ and a MAE of 1.96 μg m$^{-3}$. Similar to the MLR validation testing, LOGOCV for SSC (Table S7) produced similar metrics compared with the models using the

entire dataset. LOSOCV for SSC showed improved performance on average compared with the same process for Model 4 (Table S8), with every state exhibiting lower error metrics than the EPA's target value ($\leq$ 7 μg m$^{-3}$) for RMSE. Thus, the cluster-based models may be valid for any state in the study area.

The previous studies (McFarlane et al., 2021; Raheja et al., 2023) using an MBC to calibrate low-cost sensors are consistent

with our SSC results with lower MAEs/RMSEs for their GMR-based model compared with their MLR, indicating that an MBC is superior to an MLR approach. McFarlane et al. (2021) in their studies found for their GMR model a MAE of 0.5 less for their MLR of 2.2 μg m$^{-3}$. Similarly, Raheja et al. (2023), for their GMR model using PurpleAir sensors, found a MAE of 1.93 μg m$^{-3}$ and a RMSE of 2.58 μg m$^{-3}$, corresponding to 0.17 μg m$^{-3}$ and 0.30 μg m$^{-3}$ less than their MLR model respectively.

However, because of transferability (Raheja et al., 2023) constraints with GMR-based models, Raheja et al. (2023),
recommended to use their MLR model for future applications, although they obtained an improved model using the GMR.

Table 2: Semi-supervised clustering model development (model fit with hourly data) and application of the hourly model to daily data.

| Parameters | | Model fit with hourly data | | | | Model fit to daily data | | | |
|---|---|---|---|---|---|---|---|---|---|
| Clusters (Number of observations) | Models | $R^2$ (%) | RMSE ($\mu g\ m^{-3}$) | MAE ($\mu g\ m^{-3}$) | R (%) | $R^2$ (%) | RMSE ($\mu g\ m^{-3}$) | MAE ($\mu g\ m^{-3}$) | R (%) |
| RH $\leq$ 50 (59405) | $2.738732 + 0.425834\ PA_i - 0.008944\ RH_i + 0.079210\ T_i$ | 71 | 2.96 | 1.86 | 84 | 88 | 2.04 | 1.46 | 94 |
| RH >50 (100243) | $7.230374 + 0.412683\ PA_i - 0.085278\ RH_i + 0.070655\ T_i$ | 74 | 2.92 | 2.02 | 86 | 73 | 2.33 | 1.68 | 85 |

We compared our results with three nonlinear models that were previously tested for PurpleAir sensors. Two of these studies were not fit with data for our warm-humid climate zone study area. Malings et al. (2020) developed a two-piecewise linear model based on a threshold of 20 $\mu g\ m^{-3}$ $PM_{2.5}$ concentrations using 11 PurpleAir sensors at 2 sites in Pittsburgh. The Malings et al. (2020) paper includes DP as one of the predictors (Table 3), which violates the assumption of predictor variable independence in the correction model since a high correlation was found between DP and T. Performance metrics for the
Malings et al. (2020) model were inferior to those for our models and for the models developed by other authors (Table 3). Wallace et al. (2021, 2022) estimated correction factors based on the ratio of the mean AQS to the mean PurpleAir for all pairs of PurpleAir/AQS sites from California (Wallace et al., 2021), and from California, Washington and Oregon (Wallace et al., 2022) in separate models. Using the correction factor of 3 (ALT-CF3) recommended in Wallace et al. (2021), we calculated higher MAE and RMSE (Table 3) than for any of our models and for the Barkjohn model. Similarly, the correction model
developed by Nilson et al., (2022) to the cf=Atm data (same type of data used in their model) yielded similar $R^2$ and even higher RMSE and MAE than found with the ALT-CF3 model (Table S9). (Nilson et al., (2022) used 35 PurpleAir/FEM sites in the U.S. and Canada including 2 sites in our study area.

Table 3: Other previously developed nonlinear correction models

| Correction models | | Model fit with hourly data | | | |
|---|---|---|---|---|---|
| | | $R^2$ | RMSE | MAE | R |

| | | (%) | ($\mu g\ m^{-3}$) | ($\mu g\ m^{-3}$) | (%) |
|---|---|---|---|---|---|
| **Wallace et al. (2021)** | ALT-CF3 | 68 | 3.88 | 2.86 | 82 |
| **Nilson et al. (2022)** | pm25_atm/(1 + 0.24/(100/RH − 1)) | 68 | 4.14 | 2.98 | 82 |
| **Malings et al. (2020)** | $75 + 0.60\ PA_i - 2.50\ T_i - 0.82\ RH_i + 2.9\ DP_i$ (for PA > 20 $\mu g\ m^{-3}$) <br><br> $21 + 0.43\ PA_i - 0.58\ T_i - 0.22\ RH_i + 0.73\ DP_i$ (for PA ≤ 20 $\mu g\ m^{-3}$) | 22 | 11.08 | 9.56 | 47 |


As for the MLR, the SSC hourly model was applied to the daily average dataset. Figure 6 shows the nonlinearity of our dataset with the slope varying for each cluster for the correlation between $C_{AQS}$ and $C_{PA}$. The same aggregate of datapoints seen in Fig. 4 is also observed in the SSC models, but only in Cluster 2 (Fig. 6). This may have affected the accuracy of the model (Table 1). Applying the hourly models to daily data resulted in substantial improvement with lower uncertainties in each SSC

model compared with the hourly dataset (Table 2). Compared with the fit for Model 4 from the MLR (Table 1) to daily data, we observed that Cluster 1 presented better performance metrics than Cluster 2 (Tables 1 and 2). Compared with Model Bj applied to our daily dataset in Table 1, the daily SSC models display improved results (lower RMSE and MAE) for each cluster.

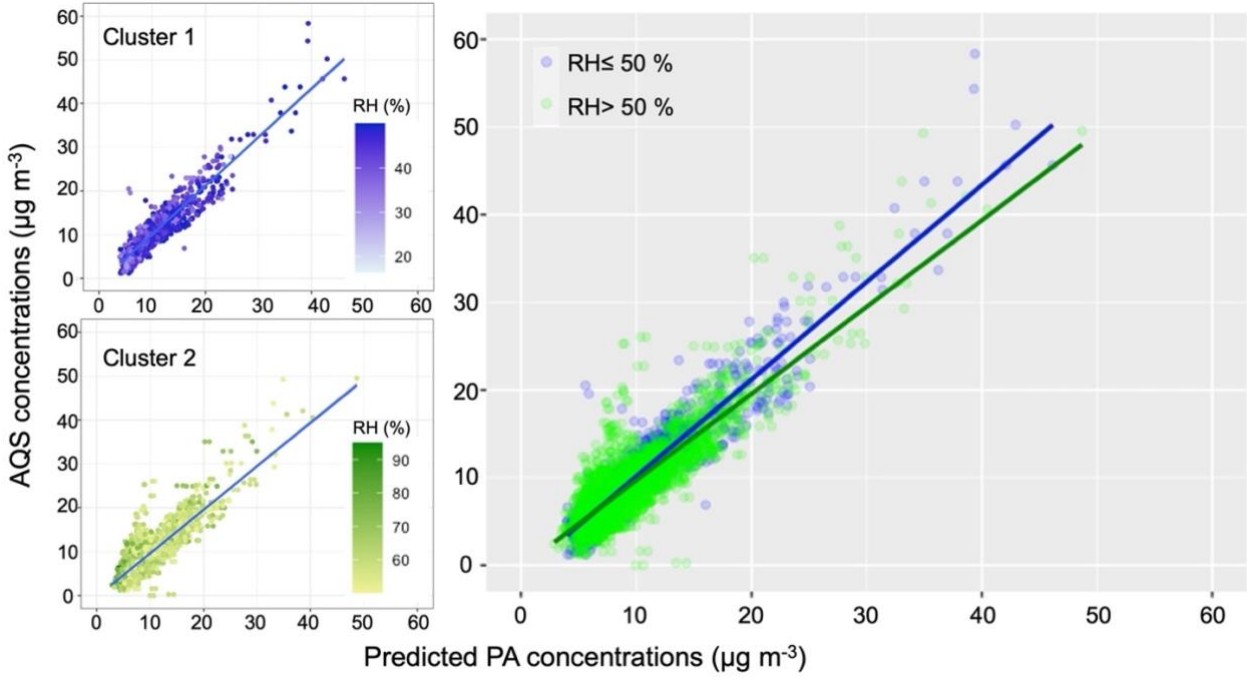


**Figure 6: Correlation between daily AQS and daily predicted PM$_{2.5}$ concentrations using the SSC model. Each cluster is presented separately on the left, and both clusters are shown on the right.**

To further assess the model performance in subgroups, Model 4 from the MLR and the SSC models were applied to daily data from 5 states of the warm-humid climate zones (Table 4). For the SSC, both models (RH $\leq$ 50 % and RH >50 %) presented good results for all the metrics compared with the hourly-data-fitted models and their application to daily data. Except for VA, where Model 4 produced lower error metric values, the SSC model outperformed the MLR for all the states.

Table 4: Application of MLR- Model 4 and SSC model to individual state. The SSC combined clusters result is the result obtained after applying each cluster to the hourly data, then added together.

| States | MLR | | | | SSC combined Clusters | | | |
|---|---|---|---|---|---|---|---|---|
| | $R^2$ (%) | RMSE ($\mu g\ m^{-3}$) | MAE ($\mu g\ m^{-3}$) | R (%) | $R^2$ (%) | RMSE ($\mu g\ m^{-3}$) | MAE ($\mu g\ m^{-3}$) | R (%) |
| SC | 56 | 3.41 | 1.92 | 75 | 57 | 3.40 | 1.87 | 75 |
| NC | 80 | 2.81 | 1.82 | 89 | 80 | 2.76 | 1.76 | 90 |
| VA | 88 | 2.70 | 2.36 | 94 | 87 | 2.77 | 2.42 | 93 |
| FL | 65 | 2.63 | 1.64 | 81 | 65 | 2.58 | 1.62 | 81 |
| TN | 75 | 3.11 | 2.21 | 87 | 75 | 3.10 | 2.19 | 87 |

**3.3 Final Model Selection**

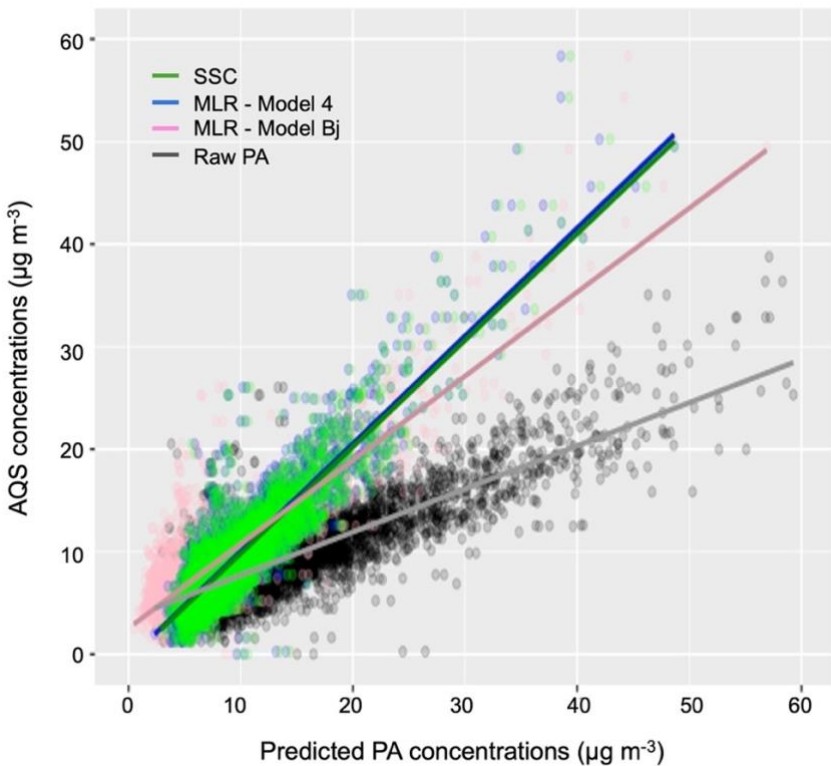

**Figure 7: Correlations and regression lines between daily AQS and daily raw/predicted PM$_{2.5}$ concentrations using the MLR, the SSC and Model Bj.**

Both Model 4 from the MLR models and the SSC models align with previous studies, producing low error and high correlation R$^2$. After comparing NOAA and PurpleAir meteorological data (Fig. S5), we included in the supplemental information (Table S10) these two sets of models (Model 4 from the MLR models and the SSC models) using NOAA meteorological data for RH and T that can be applied when meteorological information from PurpleAir sensors is biased or missing. Figure 7 summarizes the results of our study by presenting the correlation fit for the MLR (Model 4 from the MLR models), the combined clusters from the SSC, the Model Bj and the raw PurpleAir data together. Tables S11 and S12 provide an evaluation of performance of the models by Air Quality Index (AQI) categories. Our results showed that applying Model Bj to our hourly dataset improved our error metric, RMSE, of 58.73 % from the raw data. The MLR and the SSC model have lower error and higher correlation than Model Bj. A decrease of 15.91 % was obtained for RMSE from Model Bj to Model 4. However, Model 4 PM$_{2.5}$ concentrations had a higher average mean deviation (1. 99 µg m$^{-3}$) from C$_{AQS}$ than PM$_{2.5}$ concentrations from the SSC model (1.96 µg m$^{-3}$). Moreover, Model 4 PM$_{2.5}$ concentrations from the MLR models tend to be slightly higher than PM$_{2.5}$ concentrations from the SSC model at high RH and slightly lower at lower RH.

## 4. Conclusion

In conclusion, Model 4 from the MLR and the SSC model improved the error performance metrics by 16-23 % compared with the model developed by Barkjohn et al. (2021). The SSC model presented slightly better results than the overall MLR, suggesting that a clustering approach might be more accurate in areas with high humidity conditions to capture the non-linearity associated with hygroscopic growth of particles in such conditions. Therefore, the SSC model is recommended to be used for bias correction for the Southeastern United States. However, Model 4 might be an acceptable alternative for its parsimony. Applying these models to $PM_{2.5}$ PurpleAir concentrations collected in high humidity areas will help to inform communities with a high-quality estimation of their exposure. These models might also benefit communities in high humidity regions outside of the U.S. Next steps in model development may include evaluation of the transferability of these models to other humid locations in the world.

### Data availability

The processed datasets and programming codes written to perform statistical analyses and visualizations can be found on the first author's Github repository dedicated to the study : https://github.com/MartineMathieu/PurpleAir-calibration. The hourly and daily predicted concentrations are also accessible on the same repository.

All raw data can be provided by the corresponding authors upon request. PurpleAir, AQS and NOAA ISD data can be found in the following repositories:

PurpleAir data can be found using the PurpleAir API accessible on  https://api.purpleair.com and https://develop.purpleair.com

U.S. EPA AQS can be found via the AirNow API accessible on https://www.airnow.gov

NOAA ISD data can be downloaded using the R package *worldmet* or directly on https://www.ncei.noaa.gov

### Author contributions

MEM-C, AG and JR-B conceptualized the work and developed the methods. MEM-C curated the data, completed the formal analysis and figure visualizations. MEM-C and CG wrote the original draft. MEM-C, CG, AG and JR-B reviewed and edited the manuscript. JR-B acquired funding.

### Competing interests

The authors declare that they have no conflict of interest.

**Acknowledgements**

We thank the PurpleAir team for their support in obtaining PM$_{2.5}$ concentrations and meteorological PurpleAir data.

**Financial support**

This work was supported by the National Institute for Environmental Health Sciences (P42 ES013638, P30 ES025128).

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
