# Peer review of "Calibration of Low-Cost Particulate Matter Sensors PurpleAir: Model Development for Air Quality under High Relative Humidity Conditions"

_EGUsphere, 2024_

## Author Comment (AC1)

**The authors would like to thank the editor, the two reviewers, and Dr. Ouimette for their thoughtful and thorough review, and constructive remarks. We have modified the manuscript based on these comments to improve and clarify the text. Please find below detailed responses in bold blue text (with direct quotes from the revised manuscript shown in "bold, quoted and italic" text) to the comments and suggestions offered by the reviewers (shown in normal text). All line numbers in our responses correspond to the "clean" version of the revised manuscript.**

**RESPONSE TO THE COMMENTS FROM REFEREE 1**

This paper looks for a better PurpleAir correction for sensors in the US southeast. However, they only consider one equation from the literature when much additional work has been done on this topic in the past 4 years. This is not the first paper to look at nonlinear RH correction and the paper would be strengthened by comparing to other corrections in the literature that account for nonlinear RH. I have a number of other specific comments below that I hope the authors will address to strengthen their paper. The editor should also find someone to review that is more familiar with semi-supervised clustering.

**Major**

1-  I think this paper would be strengthened by considering other common corrections from the literature especially those that consider nonlinear RH terms (e.g., Wallace https://www.mdpi.com/1424-8220/22/13/4741, Nilson https://amt.copernicus.org/articles/15/3315/2022/amt-15-3315-2022.html, Malings https://www.tandfonline.com/doi/full/10.1080/02786826.2019.1623863)

**Response: The authors appreciate the reviewer's suggestion. We added a new paragraph (lines 369-377 in the Results and Discussion section) to compare the models developed in this study with other existing non-linear models as suggested. However, these models were designed for specific locations and not intended to work for a broad area. Moreover, none of these studies covered the Southeastern U.S. Malings et al. (2020) used data from 2 sites in Pittsburgh. Wallace et al. (2022) used data from California, Washington and Oregon. We added the results found by Wallace et al. (2001, 2022) and Malings et al. (2020). However, we did not include Nilson et al. (2022) since they only developed linear models using CF-1 PurpleAir data.**

**"We compared our results with nonlinear models that were previously developed and tested for PurpleAir sensor bias correction. Malings et al. (2020) developed a two-piece linear model based on a threshold of 20 µg m-3 PM2.5 concentrations using 11 PurpleAir sensors in 2 sites in Pittsburgh. The models included CPA, T, RH and DP as predictors. They found a correlation below 50 % and a MAE ranging from 3 to 5 µg m-3 (Malings et al., 2020). Some other studies (Wallace et al., 2021, 2022) estimated correction factors based on the ratio of the mean AQS to the mean PurpleAir for all pairs of PurpleAir/AQS sites first using 33 PurpleAir sensors from California (Wallace et al., 2021) and then including 182 PurpleAir sensors from**

*California, Washington and Oregon (Wallace et al., 2022). Their studies evaluated alternative PM2.5 PurpleAir estimates, however Wallace et al. (2021) also developed a correction factor for the cf_1 PM2.5 PurpleAir estimates. They calculated a range of a correction factors between 0.65 and 0.72 resulting in an overestimation of PM2.5 of 40 % compared with AQS monitors (Wallace et al., 2021)." (lines 369-377)*

2- Also, can you add a plot showing the RH nonlinearity? You say that the model shows that it shows up around 50% but where does it increase visually? Something like RH on the X axis and Sensor/Monitor on the Y axis (Examples: Zheng https://amt.copernicus.org/articles/11/4823/2018/)

**Response: We appreciate the suggestion. The plot has been added to the Supplemental Information (Figure S4) and referenced in the manuscript in line 342. Figure S4 shows the correlation between raw PM$_{2.5}$ PurpleAir concentrations and RH, with the regression line displaying the nonlinearity of PM$_{2.5}$ PurpleAir concentrations. The non-linearity curve started around RH = 50%.**

*"Figure S4 shows that non-linearity in the curve started around RH of 50%. PurpleAir datapoints that fell within a range of RH less or equal to 50% are in green and those that fell within a range greater than 50 % are shown in blue."*

[Figure]

*"Figure S4: Correlation between raw PM$_{2.5}$ PurpleAir concentrations and RH showing the nonlinearity of PM$_{2.5}$ PurpleAir concentrations. Graph a) shows all the datapoints, and graph b) is a zoom in to better display the regression line and the nonlinearity of the data."*

3- This paper discusses how the southeast is unique because it is high humidity but it would also be helpful to comment on how particle properties (e.g., composition, size distribution) are different in the south east and how that might impact the performance (e.g., Patel https://amt.copernicus.org/articles/17/1051/2024/, Jaffe https://amt.copernicus.org/articles/16/1311/2023/).

**Response:  We thank the reviewer for the suggestion. We edited the manuscript to highlight specific sources of PM₂.₅ and potential impact of particle properties in the Southeast region (lines 102-106).**

*"The high humidity condition in this part of the U.S. might affect particle composition and size distribution due to water uptake (Hagan & Kroll, 2020; Jaffe et al., 2023; Patel et al., 2024; Rueda et al., 2023). A study conducted in 2018 (Carlton et al., 2018) found large contributions (50%)  to PM₂.₅ from biogenic secondary organic aerosols (BSOA) in the Southeast U.S. region compared with the rest of the country. The elevated BSOA are attributed to heavily forested areas and large urban areas in the region (U.S. EPA, 2018; Carlton et al., 2018)." (lines 102-106)*

4- How does the recent release of the T640 correction impact this work? I agree with Dr. Ouimette that it would be helpful to list all the AQS monitors compared to, I assume some of them are Teledyne T640s.

**Response:  We thank the reviewer for the comment. We edited the manuscript to include limitations related to T640s in lines 287-289. The AQS monitors are listed in Fig. S13.**

*"Similarly, the presence of Teledyne T640s among our AQS monitors may have affected the performance of our models since positive bias of approximately 20% has been reported with T640s compared with other FEM or FRM monitors (U.S. EPA, 2024)." (lines 287-289)*

5- Were any of these sensors the alternate PMS5003s? Sear, Kaur, Kelly, https://www.sciencedirect.com/science/article/pii/S0021850223001210 How does this impact your results?

**Response:  We thank the reviewer for the comment. We edited the manuscript to include limitations related to the alternative PMS5003 in lines 290-295.**

*"Additionally, a study conducted by Searle et al. (2023) found that 12.9 % of the sensors deployed by PurpleAir between June 2021 and May 2023 reported negative bias of approximatively 3 µg m⁻³ over the long term. These PurpleAir sensors, specifically deployed between June 2021 and January 2022, and between March to May 2023 used an alternative, Plantower PMS5003 that affected the reported particle size distributions and concentrations (Searle et al., 2023). Although only 5 of our sensors, representing about 7 % of our data, fell into the reported time periods (Fig. 2), the potential presence of the alternative PMS5003 in our study may have affected the performance of our models." (lines 290-295)*

6- How much data is excluded for each of the QA methods? (AB channel comparison high, low, etc.)

**Response:  The amount of data removed at each step of the QA process was estimated, and a table with this information was added in the Supplemental Information (Table S1) and referenced in the manuscript in line 236.**

*"The QA process removed about 22 % (Table S1) of the raw data…"* **(line 236)**

*"Table S1: Percentage of hourly data removed by QA process from the initial 56 PurpleAir sensors*

| QA criteria | % removed* |
|---|---|
| Process 1: Removing NAs (PM, T, RH) | 2.026 |
| Process 2: Channels A & B agreement
    Low concentration (≤ 25 μg/m³): 537,246 obs.
    High concentration (>25 μg/m³): 80,196 obs. | 2.242
2.056 |
| Process 3: A & B concentration < 1.5 μg/m³ | 6.753 |
| Process 4: Average A & B concentration > 1000 μg/m³ | 0.005 |
| Process 5: Removing data from sensors with RH issues | 5.527 |
| Process 6:  Removing RH≠ 0-100% and T ≠ 0-130 °F | 3.484 |

*\*percent removed from the total number of observations"*

7- Figure 2 seems to show a wider range of RH with more noise over time. Is this due to seasonal differences or because the RH sensor performance is changing over time?

**Response:  We thank the reviewer for the comment. However, we did not find an appreciable difference in the RH measurements among the 3 years.**
- **2021: Mean RH of 55.07%, range of 20.20 to 80.56%**
- **2022: Mean RH of 54.31%, range of 20.37 to 89.59%**
- **2023: Mean RH of 54.91%, range of 16.43 to 95.04%**

**The wider range impression may be illustrated by the fact that January 2021 exhibited a narrower range. A shorter range was also observed for January 2022 and January 2023.**

8- Did you consider whether sensor age had any impacts on your results? (e.g., deSouza https://pubs.rsc.org/en/content/articlehtml/2023/ea/d2ea00142j)

**Response:  We thank the reviewer for this comment. The PurpleAir database did not contain information about the sensors' age or service length. We emphasize the limitations related to the sensors' age in the Results and Discussion section and how that could affect the performance of a model (lines 286-287).**

*"The model would, however, be further improved with use of newer PurpleAir sensors because, over time, the quality of the sensors degrades. This is particularly true in the hot and humid climate zone (deSouza et al., 2023)."* **(lines 286-287)**

9- "However, DP was excluded as a predictor in our study, because collinearity was found between DP, RH, and T when testing for variance inflation factor. This collinearity is attributed to the direct physical relationship between RH and T" I don't understand what this is saying? T and RH weren't significantly colinear?

**Response:** We thank the reviewer for this comment. We rephrased the paragraph to make the statement more clear (lines 166-169). RH and T were not collinear. A negative correlation of 14% was found between them. We intended to say that DP was correlated with both RH and T.

*"However, DP was excluded as a predictor in our study. DP exhibited collinearity with both RH and T when testing for variance inflation factor. In fact, a high correlation of 95% was found between DP and T. Therefore, including it would inflate the goodness of fit of the model. This result is not surprising considering the interdependent atmospheric thermodynamic relationship of DP with RH and T."* (lines 166-169)

10- Random withholding is likely not a good test of your model. It would likely be fairer to withhold by site or state. I think it isn't surprising that the model you built for your dataset is a better fit than a model built on another dataset. This is likely something to mention in the limitations.

**Response:** We appreciate the suggestion. In addition to leave one group out cross-validation (LOGOCV), which leaves out a randomly selected group, we added a leave-one-state-out cross-validation (LOSOCV) process (lines 224-226; 299-300; 357-359) which leaves out one U.S. state in the Southeast U.S. domain at a time.

*"Then, we applied a leave-one-state-out cross-validation (LOSOCV) that involves splitting the dataset into specific states to evaluate the performance of the model. In our LOSOCV, every U.S. state was left out successively and used in a validation test, while the remaining states were used to train the model."* (lines 224-226)

*"The LOSOCV resulted in a RMSE and a MAE of 3.31 µg m⁻³ and 2.29 µg m⁻³ respectively for Model 4. These values were higher than those for the LOGOCV process, which is not surprising considering the variability between states."* (lines 299-300)

*"LOSOCV for SSC showed improved performance on average compared with the same process for Model 4 (Table S8), with every state exhibiting lower error metrics than the EPA's target value (≤ 7 µg m⁻³) for RMSE."* (lines 357-359)

11- Table 2 this is interesting basically if the RH is high add 5 ug/m3 to the concentration? This difference doesn't seem to be reflected in Figure 6. Is there a typo?

**Response:** We thank the reviewer for the interesting observation. However, the difference between the 2 models did not only affect the intercept since the RH coefficient is about 10 times greater in Cluster 2 than Cluster 1. We added a sentence in the Results section to highlight the difference (lines 348-350). Moreover, the difference between the two models is not reflected in Figure 6 because Figure 6 shows the correlation between the predicted concentrations after applying the model and AQS concentrations. It would have been noticeable in a figure displaying the relationship between the raw PurpleAir data and the predicted concentrations.

*"The difference between the two models resides primarily in their intercepts and their RH coefficients (Table 2). The RH factor is 10 times greater in Cluster 2 than Cluster 1, and the intercept of Cluster 2 is about 5.5 µg m$^{-3}$ greater than Cluster 1."* **(lines 348-350)**

12- Citations should be checked for accuracy throughout see a few specific comments below.

**Response: We thank the reviewer for pointing out some errors in the citations. They have been corrected.**

13- While the results are significantly different, they are not largely different. You might consider adding evaluation of performance by AQI category to further strengthen your findings (e.g., https://www.mdpi.com/1424-8220/22/24/9669, https://amt.copernicus.org/articles/15/3315/2022/amt-15-3315-2022.html )

**Response: We appreciate the reviewer's suggestion. We presented and discussed the evaluation of performance of the models by AQI category in the Supplemental Information (Tables S11 and S12). This has been referenced in the manuscript in lines 410-411. Text describing the contents of Table S12 is also included in lines 146-150 of the Supplemental Information:**

*"Table S12 shows the total percentage of correct AQI reported by each model with their under and over estimation. Models 4 and SSC reported the highest percentage of correct AQIs with a fairly even distribution of under- and overestimation shown by SSC. Model Bj displayed a much higher underestimation than overestimation."*

*"Table S12: Summary table of the evaluation of the AQI per model for the daily dataset"*

| *Models* | *Correct AQI (%)* | *Under-estimation (%)* | *Over-estimation (%)* |
|----------|-------------------|------------------------|------------------------|
| *SSC* | *84.01* | *7.49* | *8.17* |
| *Model 4* | *84.10* | *8.70* | *6.87* |
| *Model Bj* | *83.78* | *12.81* | *3.07* |
| *Raw PA* | *72.68* | *2.99* | *26.94* |

**Minor**

14- A study conducted in 2016 (AQ-SPEC, 2016), evaluating about twelve low-cost PM2.5 sensors showed an overall good agreement between PM2.5 PurpleAir sensors and two reference monitors with a R2 of 78 % and 90 % (Wallace et al., 2021). - Is this citation correct? It seems like the beginning and ending of the sentence are citing 2 different things.

**Response: We appreciate the comment. We found this mistake in the bibliography library. The error has been corrected. See lines 42-45.**

*"A study conducted in 2016 (AQ-SPEC, 2016) to evaluate low-cost PM$_{2.5}$ sensors showed an overall good agreement between PM$_{2.5}$ PurpleAir sensors and two reference monitors with R$^2$ of 78% and 90% (AQ-SPEC, 2016). However, an overestimation of 40% was found for PurpleAir PM$_{2.5}$ concentrations compared with the reference monitors (AQ-SPEC, 2016; Wallace et al., 2021)."* **(lines 42-45)**

15- Lunden, M. M.; Parworth, C. L.; Barkjohn, K. K.; Holder, A. L.; Frederick, S. G.; Clements, A. L. Correction and Accuracy of PurpleAir PM 2.5 Measurements for Extreme Wildfire Smoke. 2022. https://doi.org/10.3390/s22249669. – This citation is incorrect

**Response: Thank you for the comment. This was again a mistake in the bibliographic library. It has been corrected to Barkjohn et al. 2022. See lines 242 and the corresponding reference in the bibliography.**

16- Line 45, 269: Why are there superscript numbers? Check for this throughout

**Response: We appreciate the comment. Line 45 was an error from a change of bibliography style and line 269 was a footnote, for which the corresponded description was missing. All the errors have been corrected. Note that these referenced line numbers are from the last draft.**

17- Figure 1: Is the number of counties by state relevant to the story you are telling?

**Response: Thank you for the comment. The table with the counties has been removed from Fig. 1.**

18- Line 270: "For all the four fitted models, average concentration of 8.80 µg m-3 , with an SD varying between 4.71- 4.84 µg m-3 were obtained, whereas Model Bj provided and a higher MAE than the four developed models with a mean of 7.67 µg m-3 and a SD of 6.08 µg m-3 ." -A little unclear if the first and second part of this sentence are comparing the same thing.

**Response: We appreciate the comment. The sentence was restated and reorganized to make the comparison more comprehensible. See lines 277-279 and lines 283-284.**

*"All four MLR-fitted models exhibited an average concentration of 8.80 µg m$^{-3}$, with a SD varying between 4.71- 4.84 µg m$^{-3}$. The Barkjohn model had a mean of 7.67 µg m$^{-3}$ and a SD of 6.08 µg m$^{-3}$."* **(lines 277-279)**

*"The Barkjohn model resulted in a higher MAE than the four models developed for this study."* **(lines 283-284)**

19- "Zheng et al. (Zheng et al., 2018) found an R2 280 value of 66 % for a 1-h averaging period after applying an MLR calibration equation to compare three PA sensors" – This is not a paper about PurpleAirs it is a paper about custom built Plantower PMS3003 sensors

**Response: We appreciate the comment. We edited the sentence to reflect the specific type of sensor used by Zheng et al. (2018), which is the same type of sensors found within PurpleAir (lines 303-304).**

*"Zheng et al. (2018), evaluating the performance of Plantower PMS3003, which is similar to the PM$_{2.5}$ sensor used in PurpleAir,…"* **(lines 303-304)**

20- I don't think R2 is usually reported as a Percentage?

**Response: Thank you for the comment. $R^2$ quantifies how much the dependent variable is determined by the independent variables, in terms of proportion of variance. Its values can be presented either in a range from 0 to 1 or in percent. See Wallace et al. (2021) as an example of $R^2$ stated in %.**

21- What is R in Table 1? Just the root of R2?

**Response: Thank you for the comment. R is the Pearson correlation. It was defined in line 264.**

22- Figure 4: I think this plot would be easier to interpret if both plots used the same color scale.

**Response: Thank you for the comment. We used different colors to differentiate between our model and the Barkjohn model. We also added the color description in the figure's caption to avoid confusion.**

23- This is a personal preference so take or leave, but I would always put the monitor on the X axis and the Sensor on the Y since the monitor is the independent variable. This is also the recommendation in the EPA performance targets.

**Response: We thank the reviewer for expressing this concern. AQS concentrations are shown on the y-axis because they are treated as the dependent variable in the model so that the PurpleAir sensor data can be adjusted accordingly.**

24- Figure 7: Is there an assumed T and RH for the lines on this plot?

**Response:** We thank the reviewer for this comment. However, there is not an assumed T or RH for the lines in the plot. T and RH are used to fit the PA data in the MLR and SSC, however. Fig. 7 is a correlation plot. We clarified in the caption:

*"Figure 7:  Correlations and regression lines between daily AQS and daily raw/predicted PM$_{2.5}$ concentrations using the MLR, the SSC and Model Bj."*

---

## Author Comment (AC2)

The authors would like to thank the editor, the two reviewers, and Dr. Ouimette for their thoughtful and thorough review, and constructive remarks. We have modified the manuscript based on these comments to improve and clarify the text. Please find below detailed responses in bold blue text (with direct quotes from the revised manuscript shown in "bold, quoted and italic" text) to the comments and suggestions offered by the reviewers (shown in normal text). All line numbers in our responses correspond to the "clean" version of the revised manuscript.

**RESPONSE TO THE COMMENTS FROM REFEREE 2**

General comments:

This paper provided the evaluation of PurpleAir correction using the warm, humid climate zones data and aimed to improve the EEPA Barkjohn model. It provides helpful information about improved performance metrics and avoids collinearity using DP, RH and T. However, the multilinear regression has been used before. There is no significant scientific insight gained with the new parameters. Several suggestions to strengthen this paper:

Response to the general comments: The authors appreciate the reviewer general comment on the scientific insight. However, we respectfully disagree with the comment. The objective of the paper was to develop and evaluate PurpleAir bias correction models (a more accurate model) for use in areas under high humidity conditions considering the sensitivity of PurpleAir sensors to humidity. Moreover, our study evaluated the performance of MLR models and a novel semi-supervised clustering method as a model-based clusters (MBC).

*"The objective of this study is to develop and evaluate PurpleAir bias correction models for use in the warm humid climate zones (2A and 3A) of the U.S. (Antonopoulos et al., 2022). First, we tested an MLR with different combinations of predictive variables. To avoid the transferability constraints observed for the GMR, our study then tested a novel semi-supervised clustering method. We used PurpleAir data and the FRM/FEM $PM_{2.5}$ data from the EPA Air Quality System (AQS) database from January 2021 to August 2023. We tested new correction models developed for the high-humidity Southeastern region of the country and compared them with the EPA nationwide PurpleAir data correction model proposed by Barkjohn et al. (2021)."* (lines 75-80)

1- Consider other correction methods and explain what can provide the best insight of the Purpleair data.

Response: We thank the reviewer for this comment. In addition to the two methods tested in our study and their comparison with the model developed by Barkjohn et al. (2021), a paragraph was added to the manuscript to compare the results of our study with other non-linear models previously used (lines 369-377). Please see our response to Referee 1, comment #1 copied below.

**Response to Reviewer #1:** The authors appreciate the reviewer's suggestion. We added a new paragraph (lines 369-377 in the Results and Discussion section) to compare the models developed in this study with other existing non-linear models as suggested. However, these models were designed for specific locations and not intended to work for a broad area. Moreover, none of these studies covered the Southeastern U.S. Malings et al. (2020) used data from 2 sites in Pittsburgh. Wallace et al. (2022) used data from California, Washington and Oregon. We added the results found by Wallace et al. (2001, 2022) and Malings et al. (2020). However, we did not include Nilson et al. (2022) since they only developed linear models using CF-1 PurpleAir data.

*"We compared our results with nonlinear models that were previously developed and tested for PurpleAir sensor bias correction. Malings et al. (2020) developed a two-piece linear model based on a threshold of 20 µg m-3 PM2.5 concentrations using 11 PurpleAir sensors in 2 sites in Pittsburgh. The models included CPA, T, RH and DP as predictors. They found a correlation below 50 % and a MAE ranging from 3 to 5 µg m-3 (Malings et al., 2020). Some other studies (Wallace et al., 2021, 2022) estimated correction factors based on the ratio of the mean AQS to the mean PurpleAir for all pairs of PurpleAir/AQS sites first using 33 PurpleAir sensors from California (Wallace et al., 2021) and then including 182 PurpleAir sensors from California, Washington and Oregon (Wallace et al., 2022). Their studies evaluated alternative PM2.5 PurpleAir estimates, however Wallace et al. (2021) also developed a correction factor for the cf_1 PM2.5 PurpleAir estimates. They calculated a range of a correction factors between 0.65 and 0.72 resulting in an overestimation of PM2.5 of 40 % compared with AQS monitors (Wallace et al., 2021)."* (lines 369-377)

2- Typically, the low-cost sensors measure the PM base on the optical size, and it is unclear how they can accurately predict the aerodynamic size and get the correct PM2.5. The conversion of particle aerodynamic size to optical size, or vice versa, is not straightforward because it depends on several factors, including the particle's shape, density, and refractive index. Are the FRM/FEM monitors filter-based measurements? How does the linear regression provide reliable information?

**Response:** We thank the reviewer for the comment. FRM/FEM monitors are reference-grade monitors designated by EPA. EPA has evaluated every FRM/FEM to ensure that it is producing accurate concentrations based specific standards (40 CFR Appendix L to Part 50). Moreover, we have already pointed out in the manuscript that optical sensors have many challenges in accurate detection of PM2.5 (lines 47-50).

*"Most low-cost PM sensors, including the PurpleAir sensor, utilize optical sensors based on the light-scattering principle to estimate PM mass concentration. Thus, they are subject to measurement errors from various factors, including particle size, composition, optical properties, and interactions of particles with atmospheric water vapor (Hagan & Kroll, 2020; Rueda et al., 2023; Zheng et al., 2018; Zusman et al., 2020)."* (lines 47-50)

With regard to the reliability of the modeling method, the linear regression is designed to correct less accurate PA sensors based on the more accurate AQS monitors. The performance of a linear regression is measured in general by its precision of linearity using $R^2$ and R and by the accuracy of the error metrics. The performance metrics evaluated in our study are presented in lines 210-217.

**Specific comments:**

3- Line 127-129, Please explain how to determine the detection range for PurpleAir? The reference used 1.15-2.55? This paper used 1.5? Why not 1.6? or 1.75?

**Response: We appreciate the comment. One of the references was missing. We clarified the statement and added the missing reference in lines 134-136.**

*"We applied a series of data exclusion criteria for quality control. First, we used a detection limit of 1.5 $\mu g\ m^{-3}$ for the PurpleAir data. This value is equivalent to the average of the values reported by Tryner et al. (2020) and Wallace et al. (2021) for the cf_1 data series." (lines 134-136)*

4- Line 131, What is the difference between the two channels? Should we expect them to agree in a certain percentage at low and high concentrations?

**Response: We thank the reviewer for expressing the concern. There is no difference between the design of the 2 channels. They are both PM2.5 sensors arbitrarily designated as Channels A and B. We edited the sentence to add the word *"arbitrarily"* for more clarity (line 138).**

**The data cleaning criteria for the agreement between the 2 channels for both low and high concentrations are already defined in the manuscript in lines 136-146.**

5- Line 141, For each site, how much data remained? Does this data cleaning cause any bias in the data collection?

**Response: We thank the reviewer for the comment. Fig. S1 presents the number of data points remaining to be used in the study per site (n from Fig. S1 corresponds to the number of data points per PurpleAir site). Moreover, we added a table (Table S1) to present how much data were removed in the process at each step.**

**The role of the data cleaning is to minimize biases in the modeling process.**

*"The QA process removed about 22 % (Table S1) of the raw data…" (line 236)*

*"Table S1: Percentage of hourly data removed by QA process from the initial 56 PurpleAir sensors*

| QA criteria | % removed* |
|---|---|
| *Process 1: Removing NAs (PM, T, RH)* | *2.026* |
| *Process 2: Channels A & B agreement* | |
| *Low concentration (≤ 25 μg/m³): 537,246 obs.* | *2.242* |
| *High concentration (>25 μg/m³): 80,196 obs.* | *2.056* |
| *Process 3: A & B concentration < 1.5 μg/m³* | *6.753* |
| *Process 4: Average A & B concentration > 1000 μg/m³* | *0.005* |
| *Process 5: Removing data from sensors with RH issues* | *5.527* |
| *Process 6: Removing RH≠0-100% and T ≠0-130 °F* | *3.484* |

*\*percent removed from the total number of observations"*

6- Section 2.4.2, the equations are confusing. Will the beta 2 in equation 2 be the same as the beta 2 in equation 3?

**Response: We thank the reviewer for expressing the concern. The equations follow the general mathematical notation of a multilinear regression model (see equation below). Each beta is the regression coefficient of a predictor X, whose name is defined in the equation ($C_{PA}$: PurpleAir $PM_{2.5}$ concentration, RH: relative humidity, T: temperature). They will not have the same values.**

$$Y = \beta_0 + \beta_1 X_1 + \beta_2 X_2 + \ldots + \beta_p X_p + \varepsilon$$

7- Table 1, the parameters from each model have a very high precision. Is it realistic to include such high precision?

**Response: We thank the reviewer for the comment. We included such a high precision with many significant figures so that users of our models would not have rounding errors in their datasets.**

8- Figure 4 the data plotted seemed to be from two groups. One follows 1:1 line, and the other one follows 2:1 line. Cluster 2 still has the 2:1 group. Is there any other reason for this 2:1 group?

**Response: We thank the reviewer for the interesting observation. We acknowledge that this represents an area of uncertainty. We added a sentence in the Results and Discussion section (lines 322-324 and lines 383-385) to acknowledge the cluster formation as a limitation.**

*"An aggregate of datapoints can be seen on the left-hand side of the correlation plots (Fig. 4) to deviate from the model fit line. These data probably influenced the performance metrics of the models."* **(lines 322-324)**

*"The same aggregate of datapoints seen in Fig. 4 is also observed in the SSC models, but only in Cluster 2 (Fig. 6). This may have affected the accuracy of the model (Table 1)."* **(lines 383-385)**

---

## Author Comment (AC3)

**The authors would like to thank the editor, the two reviewers, and Dr. Ouimette for their thoughtful and thorough review, and constructive remarks. We have modified the manuscript based on these comments to improve and clarify the text. Please find below detailed responses in bold blue text (with direct quotes from the revised manuscript shown in "bold, quoted and italic" text) to the comments and suggestions offered by the reviewers (shown in normal text). All line numbers in our responses correspond to the "clean" version of the revised manuscript.**

**RESPONSE TO THE COMMENTS FROM JAMES OUIMETTE, 09 May 2024**

Hi,
Thank you for your preprint. I have a couple suggestions that could improve your paper.
Could you please provide a table with the following information about each of the PA sensors used in this study:
PurpleAir ID number; AQS number for the regulatory monitoring site; name of regulatory PM2.5 monitor (e.g., Teledyne T640x, Met One BAM 1020, etc); distance from PurpleAir to regulatory PM2.5 monitor; name of the NOAA site used for relative humidity and temperature data; distance from PurpleAir to NOAA site.

**Response: Thank you for the comment. A table (Table S13) with the suggested information has been added in the Supplemental Information.**

*"Table S13: List of the PurpleAir sensors and Federal Reference Methos (FRM) or Federal Equivalence Method (FEM) used in the study with the estimated distance between stations"*

| Site # | PA ID | AQS ID | FRM or FEM Type | Distance PA-AQS (km) | NOAA ID | Distance PA-NOAA (km) |
|--------|-------|--------|-----------------|----------------------|---------|------------------------|
| FL | 25949 | 121150013 | Teledyne T640 | 0.028 | 722115-12871 | 13.392 |
| FL | 16317 | 121150013 | Teledyne T640 | 0.123 | 722115-12871 | 13.350 |
| FL | 101259 | 120570113 | Teledyne T640 | 0.011 | 722110-12842 | 7.877 |
| FL | 149710 | 120570113 | Teledyne T640 | 0.011 | 722110-12842 | 7.874 |
| *GA | 142428 | 131210056 | Met One BC-1060 | 0.500 | 722190-13874 | 17.434 |
| *GA | 148123 | 131210056 | Met One BC-1060 | 0.500 | 722190-13874 | 17.434 |
| SC | 35139 | 450190020 | Teledyne T640X | 0.438 | 722080-13880 | 10.972 |
| NC | 98623 | 371190041 | Met One BAM-1020 | 0.307 | 723140-13881 | 18.780 |
| NC | 6008 | 370670022 | Teledyne T640X | 0.005 | 723193-93807 | 2.445 |
| VA | 178279 | 518100008 | Teledyne T640X | 0.052 | 723080-13737 | 7.038 |
| TX | 166421 | 482010046 | Met One BAM-1022 | 0.053 | 720594-00188 | 16.597 |
| TN | 176311 | 470450004 | Met One BAM-1022 | 0.033 | 723347-03809 | 6.604 |
| TN | 93593 | 471130010 | Met One BAM-1022 | 0.066 | 723346-03811 | 16.645 |
| TN | 51741 | 470990003 | Met One BAM-1022 | 0.004 | 723235-13896 | 46.322 |
| TN | 51867 | 470990003 | Met One BAM-1022 | 0.001 | 723235-13896 | 46.323 |
| *TN | 51737 | 470990003 | Met One BAM-1022 | 0.002 | 723235-13896 | 46.321 |
| TN | 93577 | 471192007 | Met One BAM-1022 | 0.086 | 723249-00463 | 21.910 |

| TN | 93645 | 470370023 | Teledyne T640X | 0.064 | 723270-13897 | 9.235 |
|----|-------|-----------|----------------|-------|--------------|-------|
| TN | 51921 | 470370023 | Teledyne T640X | 0.058 | 723270-13897 | 9.264 |
| TN | 51873 | 470370023 | Teledyne T640X | 0.076 | 723270-13897 | 9.262 |
| TN | 116559 | 470370023 | Teledyne T640X | 0.474 | 723270-13897 | 9.589 |

*\* sensor removed after QA process*

The sites that you chose are characterized by high dew points, resulting from both high RH and high temperatures.

Your graphs comparing RH between the PurpleAir and its corresponding NOAA site is inadequate for assessing whether or not the NOAA site is representative. The best way to show if the PurpleAir and its corresponding NOAA site are sampling similar air is to compare their hourly average dew points. That's because the PurpleAir slightly heats the air sample, resulting in a higher temperature and lower RH compared to the NOAA site. However, the water content and dew point should be the same for the PurpleAir and the NOAA site.

Could you please provide graphs comparing the hourly average dew points for your 21 sites.

**Response: Thank you for the comment. We included a comparison section between DP from NOAA sites and PurpleAir in the Supplemental Information (Fig. S5, see below) and referenced in line 406. However, we wanted to point out that DP was excluded from our study because DP exhibited correlation with both RH and T in the regression analysis when testing for variance inflation factor. A high correlation of 95% was found between DP and T. Therefore, including it would inflate the goodness of fit of the model.**

*"After comparing NOAA and PurpleAir meteorological data (Fig. S5), we included ...."* (line 406)

*"To better estimate if NOAA meteorological data can replace PurpleAir meteorological data, we compared their DP since the water content and DP should be the same for the PurpleAir and the NOAA sites. Figure S5, which used all hourly datapoints of our study, showed a Pearson correlation of 96%. Except TX, which represented only 0.32% of our dataset and exhibited a low correlation (13%), all the NOAA sites resulted in a high correlation ranging from 80 to 97% with PurpleAir sites."* (Lines 127-131 of the Supplemental Information)

[Figure]

*Figure S5: Correlation between DP from PurpleAir and NOAA*

Thanks,
Jim Ouimette

---

## Author Response (AR2)

**The authors thank the reviewer for their second set of comments. We have modified the manuscript based on these comments to improve and clarify the text. Please find below our detailed responses in bold blue text (with direct quotes from the revised manuscript shown in "bold, quoted and italic" text). The reviewer's comments are shown in black unformatted text. All line numbers in our responses correspond to the "clean" version of the revised manuscript.**

**RESPONSE TO THE COMMENTS FROM THE REVIEWER**

While some improvements have been made to this paper, in my opinion the authors have not yet adequately shown what their work adds to past work. They use a new correction method but do not compare to the accuracy that could have been achieved with other commonly used corrections in the literature. They only compare to the Barkjohn equation. The Barkjohn equation is one of the older PurpleAir corrections and multiple other corrections showing better performance than the Barkjohn equation have been published in the past 3 years (e.g., Wallace, Nilson). This was an issue I brought up on the last round of the draft that the authors did not address. I hope that the authors will address this comment along with my other specific comments below. They seem to have added additional inaccuracies in some places and may have errors in some figures. I do not feel this paper is ready for publication but think that it could be after another round of major revisions.

Major:
1- On the last round of revisions, I and the other reviewer requested that the author consider other corrections common in the literature. To me considering other equations would mean applying corrections of similar form to their dataset and then comparing how the coefficients and performance compare to past work and other corrections. Instead, they have just added a paragraph summarizing the results from the past studies.

**Response: We thank the reviewer for the comment. We edited the previous paragraph (lines 377-389) to apply other existing non-linear correction models to our PurpleAir sensors data and compare them with our developed models.**

**"**
**We compared our results with three nonlinear models that were previously tested for PurpleAir sensors. Two of these studies were not fit with data for our warm-humid climate zone study area. Malings et al. (2020) developed a two-piecewise linear model based on a threshold of 20 μg m⁻³ PM₂.₅ concentrations using 11 PurpleAir sensors at 2 sites in Pittsburgh. The Malings et al. (2020) paper includes DP as one of the predictors (Table 3), which violates the assumption of predictor variable independence in the correction model since a high correlation was found between DP and T. Performance metrics for the Malings et al. (2020) model were inferior to those for our models and for the models developed by other authors (Table 3). Wallace et al. (2021, 2022) estimated correction factors based on the ratio of the mean AQS to the mean PurpleAir for all pairs of PurpleAir/AQS sites from California (Wallace et al., 2021), and from California, Washington and Oregon (Wallace et al., 2022) in separate models. Using the correction factor of 3 (ALT-CF3) recommended in Wallace et al.**

*(2021), we calculated higher MAE and RMSE (Table 3) than for any of our models and for the Barkjohn model. Similarly, the correction model developed by Nilson et al. (2022) to the cf=Atm data (same type of data used in their model) yielded similar $R^2$ and even higher RMSE and MAE than found with the ALT-CF3 model (Table S9).  Nilson et al. (2022) used 35 PurpleAir/FEM sites in the U.S. and Canada including 2 sites in our study area.*

*Table 3: Other previously developed nonlinear correction models*

| Correction models | | Model fit with hourly data | | | |
|---|---|---|---|---|---|
| | | $R^2$ (%) | RMSE ($\mu g\ m^{-3}$) | MAE ($\mu g\ m^{-3}$) | R (%) |
| *Wallace et al. (2021)* | *ALT-CF3* | 68 | 3.88 | 2.86 | 82 |
| *Nilson et al. (2022)* | $pm25\_atm/(1 + 0.24/(100/RH - 1))$ | 68 | 4.14 | 2.98 | 82 |
| *Malings et al. (2020)* | $75 + 0.60\ PA_i - 2.50\ T_i - 0.82\ RH_i + 2.9\ DP_i\ (for\ PA > 20\ \mu g\ m^{-3})$ $21 + 0.43\ PA_i - 0.58\ T_i - 0.22\ RH_i + 0.73\ DP_i\ (for\ PA \le 20\ \mu g\ m^{-3})$ | 22 | 11.08 | 9.56 | 47 |

"

2- Met One BC-1060 – this is a black carbon monitor not a PM2.5 monitor. Why would you compare it to PM2.5? It looks like this site also runs an FRM R & P Model 2025 PM-2.5 Sequential Air Sampler w/VSCC that would have been more appropriate to compare to (https://www.epa.gov/outdoor-air-quality-data/interactive-map-air-quality-monitors). I don't think these sites should be included in Figure 1. It is misleading when you didn't have any PM2.5 data you compared to in GA to include it in the states that you had comparisons for. At a minimum it should be in a different color if not included in the model development.

**Response: Thank you for pointing that out. We double checked the monitor type. The monitor in GA mentioned in our study is indeed the R & P Model 2025 PM-2.5 Sequential Air Sampler w/VSCC. The type of monitor has been corrected in Table S13. We apologize for the error.**

3-  If you are using only hourly data, then you will never use FRM data since it is only 24-hr averages. You should clarify this throughout including "included only hourly average PurpleAir data points that had a spatial (within the calculated radius) correspondence to hourly FRM or FEM concentration". When you do the 24-hr analysis do you only use 1-hr data averaged up to 24-hr data or do you also pull in the 24-hr FRM data?

**Response: Thank you for the comment. All of our AQS data, including the averaged 24-hr data, were initially hourly data. To address the comment, we edited the manuscript to add that detail (Footnote 1) and remove the "FRM" acronym in that statement (lines 150-152).**

*"Following data cleaning, the final PurpleAir concentration ($C_{PA}$) dataset used in our study was obtained by averaging Channels A and B and included only hourly average PurpleAir data points that had a spatial (within the calculated radius) correspondence to hourly FEM[1] concentration ($C_{AQS}$) data."*

*"Footnote 1: The AQS reference monitors used in our study were FEM monitors."*

4- Figure 4: I am surprised that Model 4 performs better. It seems like the green plot shows more scatter and also has a lower slope further from the 1:1 line. I also don't see any negative data from the Barkjohn equation even though you stated that was an issue. Comparing Figure 4 to 7, I think one of these plots is wrong in Figure 7 model Bj is closer to the y axis while in Figure 4 Model 4 is closer to the Y axis. Am I missing something or is there a mistake in your figure labeling?

**Response: We are very grateful to you for identifying this error. Upon further review, we found that the captions from Figure 4a and Figure 4b were interchanged. We edited the caption to correct the mistake (lines 332-334).**

*"Figure 4: Positive linear correlation between daily AQS and daily predicted PM$_{2.5}$ concentrations with RH distribution (a) AQS and predicted PM$_{2.5}$ concentrations using Model 4 of the MLR process shown in purple (b) AQS and predicted PM$_{2.5}$ concentrations using the Barkjohn model shown in green."*

Minor:
5- "However, we did not include Nilson et al. (2022) since they only developed linear models using CF-1 PurpleAir data." Their work was based on the cf_atm data based on the corrigendum they released last July https://amt.copernicus.org/articles/15/3315/2022/amt-15-3315-2022-corrigendum.pdf.

**Response: Nilson et al. (2022) has now been included. Please see our response to comment #1.**

6- Figure S4. It is hard to understand what this means since you haven't normalized for the true monitor concentration (i.e., dividing each hourly PurpleAir concentration by that hours monitor concentration)

**Response: Thank you for the comment. Figure S4 has been edited to normalize the data.**

"
* * *
[1] The AQS reference monitors used in our study were FEM monitors.

[Figure]

*Figure S1: Correlation between the ratio of raw PM₂.₅ PurpleAir and AQS concentrations and RH showing the nonlinearity of PM₂.₅ PurpleAir concentrations. Graph a) represents the entire dataset, and graph b) is a zoom in to better display the regression line and the nonlinearity of the data."*

7- The authors didn't check their dataset to see if there were any alternative PMS5003s they just said that could have impacted their results when they easily could have checked the ratio between the bin data to see if they had any of the alternative PMS5003s.

**Response: Thank you for the comment. We edited the previous paragraph to specify how many alternative PMS5003s were found among our sensors (lines 294-299).**

*"Additionally, a study conducted by Searle et al. (2023) found that 12.9 % of the sensors deployed by PurpleAir between June 2021 and May 2023 reported negative bias of approximatively 3 µg m-3. These PurpleAir sensors, specifically deployed between June 2021 and January 2022 and between March to May 2023, used an alternative Plantower PMS5003, which affected the reported particle size distributions and concentrations (Searle et al., 2023). Based on the technique developed by Searle et al. (2023) to identify PMS5003 sensors, we estimated that only one of our sensors (sensor ID: 116559), representing 0.62% of our data, fell into this category. This may have a slight effect on the performance of our models."*

8- "The LOSOCV resulted in a RMSE and a MAE of 3.31 µg m-3 and 2.29 µg m-3 respectively for Model 4. These values were higher than those for the LOGOCV process, which is not surprising considering the variability between states." 3.32 is the value in table S8.

**Response: Thank you for the comment. The typo has been corrected in the manuscript (lines 303-304).**

*"The LOSOCV resulted in a RMSE and a MAE of 3.32 μg m$^{-3}$ and 2.29 μg m$^{-3}$ respectively for Model 4."*

9- "To ensure data accuracy, AQS data are collected by FRM or FEM, which are typically filter-based monitors (U.S. EPA, 2023b)" I'm not sure this is helpful. While FRM are filter based and a beta attenuation monitor uses filter tape, the T640 is optical and is the most common FEM used in your study.

**Response: Thank you for the comment. We edited the manuscript to remove the filter-based statement (line 127).**

*"To ensure data accuracy, AQS data are collected by FRM or FEM (U.S. EPA, 2023b)."*

10- "Based on the electronic effects of water uptake" Is this accurate? I thought this was due to the particles up taking water.

**Response: Thank you for the comment. We edited the manuscript and restated the sentence (lines 164-166).**

*"Because measurement errors are related to water uptake by particles (Hagan & Kroll, 2020; Rueda et al., 2023; Wallace et al., 2021), temperature (T) and relative humidity (RH) are the most commonly found bias correction parameters in the literature (Ardon-Dryer et al., 2020; Bi et al., 2020; Magi et al., 2020; Malings et al., 2020; Wallace et al., 2021) for the PurpleAir."*

11- Table S13 – it would be helpful to include the number of valid data points for each pair to better understand what part of the time period is represented since there is so much missing data according to Tables S1.

**Response: Thank you for the comment. The number of valid data points for each pair of PurpleAir/AQS has been added to Table S13. Figure 2, produced after the QA process, also shows what time period is represented by each pair of data.**

12- Line 383 (track changes version): as I stated in my first review Zheng et al. is not about PurpleAir sensors PurpleAir should be replaced with Plantower in this sentence.

**Response: We appreciate the comment. We edited the sentence to replace "PurpleAir" with "Plantower." (lines 308-311)**

*"Zheng et al. (2018), evaluating the performance of Plantower PMS3003, which is similar to the PM$_{2.5}$ sensor used in PurpleAir, found an R$^2$ value of 66 % for a 1-h averaging period after applying an MLR calibration equation to compare three Plantower sensors against each other and a co-located reference monitor over a period of 30 days."*

13- Line 421: SD is not listed in Table 1. Is it an error metric?

**Response: We appreciate the comment. The sentence was restated to remove the confusion (lines 329-330).**

*"An evaluation of Model Bj applied to our warm-humid climate zone daily PurpleAir datasets revealed substantially higher error metrics than the other models (Table 1)."*

14- Line 489: "We compared our results with some nonlinear models that were previously tested for PurpleAir sensors" I don't think you have done that. You haven't even drawn the conclusion for the reader on whether their error is higher or lower than yours and you didn't apply them to your dataset.

**Response: Thank you for the comment. Please see our response to comment #1.**

15- Line 465: " Click or tap here to enter text." Remove

**Response: Thank you for the comment. This erroneous text has been removed.**

---

## Author Response (AR3)

**The authors thank the editor for the comments. We have modified the manuscript based on these comments to improve and clarify the text. Please find below our detailed responses in bold blue text (with direct quotes from the revised manuscript shown in "bold, quoted and italic" text). The editor's comments are shown in black unformatted text. All line numbers in our responses correspond to the "clean" version of the revised manuscript.**

**RESPONSE TO THE COMMENTS FROM THE REVIEWER**

There are a few minor issues that need to be addressed prior to final publication.

1) One of the reviewers made a last recommendation: The MLR model parameters have a significantly high precision. It will be helpful to understand the rounding errors this model causes if the parameter precision is reduced. Will the R-square decrease with parameters that have fewer precisions?

**Response: Thank you for the question. We tested the model for different levels of precision for the MLR coefficients. $R^2$ was robust to these changes. However, RMSE increased slightly as coefficient precision was decreased from 8 to 4 to 3 significant figures, while MAE increased with decreased precision for Model 3. Changes in these parameters were at the hundredth place. Please see Table S14 in the Supplemental Material.**

*"Table S14: Tests of model coefficient precision*

| *Number of significant figures* | *RMSE* | *MAE* | *$R^2$* |
|---|---|---|---|
| *Model 3* | | | |
| *8* | *2.318026* | *1.674111* | *0.7717575* |
| *4* | *2.318178* | *1.674819* | *0.7717097* |
| *3* | *2.320251* | *1.664604* | *0.7724541* |
| *Model 4* | | | |
| *8* | *2.236673* | *1.595438* | *0.7871297* |
| *4* | *2.237311* | *1.597079* | *0.7870842* |
| *3* | *2.244286* | *1.610342* | *0.7865547* |

*"*

2) Please label the sites from Fig 2 in Fig 1. Or at least provide a figure/table in the SI that defines the sites and their metadata (lat, lon, elevation, etc.). Also, define PA in the caption of Fig 1, although most can figure out this is PurpleAir.

**Response: Thank you for the comment. We added the corresponding longitude and latitude to each PurpleAir to Table S13 in the Supplemental Material.**

3) Lines 155-157 - Which NOAA database? Add a better descriptor and perhaps a URL.

**Response: Thank you for the comment. We edited the text to add the specific NOAA database. The following sentence provides the reference of the database (lines 156-158).**

*"We compared hourly RH from the PurpleAir with the corresponding hourly RH from the National Oceanic and Atmospheric Administration (NOAA) Integrated Surface Database (ISD). The NOAA data were downloaded using the R package worldmet (worldmet: Import Surface Meteorological Data from NOAA ISD)."*

4) Section 2.4.2 - It is not clear where the other models came from. Based on a statement farther down, it appears they were developed/proposed by the authors. Please provide some more details as to how these were formulated. Also, please define all parameters that were not defined in the section above (i.e., the betas and epsilon).

**Response: Thank you for the comment. We edited the text to provide more details about the models (lines 180-181) and to define the parameters (lines 187-188).**

*"Based on the evaluated predictors, we developed Model 1-4. The four proposed models and the Barkjohn model were structured as follows:"*

*"For each model, $\beta_0$ represents the intercept, $\beta_1$ - $\beta_3$ are the coefficient of the predictors $C_{PA}$, RH and T respectively, and $\varepsilon$ is the error term."*

5) Fig 2 - I assume the axes are 0-1 but that needs to be defined in the caption for clarity. Also, SD should be shown in panel b, perhaps as a shaded area, so the variability in the measurements is clear.

**Response: Thank you for the comment. We edited the text to clarify the y-axis, and we added the SD for RH in the caption (lines 257-259).**

*"Figure 2: (a) Summary statistics and time series (yellow lines) of daily average RH for each PurpleAir site showing the presence of data (green) and missing data (red). The y axis represents RH scaled from zero to the maximum daily value. The percentage of data captured per year is also provided. (b) Time series of daily average RH for the entire dataset with a SD of 10.56 %."*

6) Lines 285-286 - Be clear that the improved performance is due to slightly reduced errors and not necessarily better correlation coefficients.

**Response: Thank you for the comment. We want to draw your attention to sentences that follow that statement. They provide detailed information about the comparison between the models in regard to their $R^2$ and error metrics. (lines 288-293).**

*"Our dataset illustrates improved predictive performance for our four MLR-fitted models compared with the Barkjohn model (Table 1). The Barkjohn model presented a higher $R^2$, as a measure of the goodness of fit, than Model 1, however Model 1 is improved with respect to all error metrics. The Barkjohn model resulted in a higher MAE than the four models developed*

*for this study. The best model fit was observed for Model 4, incorporating C$_{PA}$, T, and RH, with substantially better prediction performance metrics compared with the other models (Table 1)."*

7) Lines 360-361 - What is the odd text? "Click here or tap to enter text." I assume some sort of error…?

**Response: Thank you for the comment. This erroneous text has been removed.**

8) Table 4. Please provide the R^2 values since they are provided elsewhere.

**Response: Thank you for the comment. We edited the table to provide the R$^2$ values.**

"

*Table 4: Application of MLR- Model 4 and SSC model to individual state. The SSC combined clusters result is the result obtained after applying each cluster to the hourly data, then added together.*

| States | MLR | | | | SSC combined Clusters | | | |
|---|---|---|---|---|---|---|---|---|
| | $R^2$ (%) | RMSE ($\mu g\ m^{-3}$) | MAE ($\mu g\ m^{-3}$) | R (%) | $R^2$ (%) | RMSE ($\mu g\ m^{-3}$) | MAE ($\mu g\ m^{-3}$) | R (%) |
| SC | 56 | 3.41 | 1.92 | 75 | 57 | 3.40 | 1.87 | 75 |
| NC | 80 | 2.81 | 1.82 | 89 | 80 | 2.76 | 1.76 | 90 |
| VA | 88 | 2.70 | 2.36 | 94 | 87 | 2.77 | 2.42 | 93 |
| FL | 65 | 2.63 | 1.64 | 81 | 65 | 2.58 | 1.62 | 81 |
| TN | 75 | 3.11 | 2.21 | 87 | 75 | 3.10 | 2.19 | 87 |

"

---

## Author Response (AR4)

**The authors thank the editor for the comments. We have modified the manuscript based on these comments to improve and clarify the text. Please find below our detailed responses in bold blue text (with direct quotes from the revised manuscript shown in "bold, quoted and italic" text). The editor's comments are shown in black unformatted text. All line numbers in our responses correspond to the "clean" version of the revised manuscript.**

**RESPONSE TO THE COMMENTS FROM THE REVIEWER**

In the data availability statement, please provide the access URLs to all data used in the manuscript (i.e., PurpleAir, AQS, NOAA ISD, etc.).

Will the model output data (i.e., predicted PA concentration data in figures 4, 6, and 7) be made publicly available anywhere? Please see the AMT data policy here: https://www.atmospheric-measurement-techniques.net/policies/data_policy.html.

**Response: Thank you for the comments. We updated the "Data availability statement" to provide the link to the data used in our study. The model output data (Model 4, Model Bj and SSC model) are also available via the same link (lines 444- 453).**

*"Data availability*

*The processed datasets and programming codes written to perform statistical analyses and visualizations can be found on the first author's Github repository dedicated to the study : https://github.com/MartineMathieu/PurpleAir-calibration . The hourly and daily predicted concentrations are also accessible on the same repository.*

*All raw data can be provided by the corresponding authors upon request. PurpleAir, AQS and NOAA ISD data can be found in the following repositories:*
*PurpleAir data can be found using the PurpleAir API accessible on https://api.purpleair.com and https://develop.purpleair.com*
*U.S. EPA AQS can be found via the AirNow API accessible on https://www.airnow.gov*
*NOAA ISD data can be downloaded using the R package worldmet or directly on https://www.ncei.noaa.gov "*